# Understanding Severe Winter Haze Events in the North China Plain in 2014: Roles of Climate Anomalies

Zhicong Yin[12], Huijun Wang[123], Huopo Chen[231]

[1] Key Laboratory of Meteorological Disaster, Ministry of Education / Joint International Research Laboratory of Climate and Environment Change (ILCEC) / Collaborative Innovation Center on Forecast and Evaluation of Meteorological Disasters (CIC-FEMD), Nanjing University of Information Science & Technology, Nanjing 210044, China

[2]Nansen-Zhu International Research Centre, Institute of Atmospheric Physics, Chinese Academy of Sciences, Beijing, China

[3]Climate Change Research Center, Chinese Academy of Sciences, Beijing, China

*Correspondence to*: Zhicong Yin (yinzhc@163.com)

**Abstract.** Atmospheric pollution has become a serious environmental and social problem in China. The number of winter (December–February) haze days over the North China Plain ($WHD_{NCP}$) was largest in 2014 during the past 30 years. In addition to anthropogenic influence, the roles of climate anomalies were also vital. Thus, it is necessary to analyse the anomalous atmosphere circulations associated with haze pollution of this year in detail. Near surface, the weaker East Asian winter monsoon pattern, inducing southerly over North China Plain, could aggravate the situation of haze. In the lower and middle troposphere, taking the anti-cyclone circulation over North China as an intermedia, the positive phases of the East Atlantic/West Russia (EA/WR), the Western Pacific (WP) and the Eurasia (EU) patterns led to a worse air pollution dispersion condition that contributed to a larger number of $WHD_{NCP}$. In 2014, these three patterns could be recognized from the wind anomalies in the lower troposphere. The preceding autumn (September–November) Arctic sea ice (ASI) anomalies over the Eastern Hemisphere and the warmer winter surface over Eurasia might have induced or intensified the positive EA/WR pattern in 2014. These two external forcings, together with the pre-autumn sea surface temperature anomalies in the Pacific, might have also stimulated or enhanced the positive EU-like patterns. The anomalous surface temperature in autumn 2014 was efficient in intensifying anomalous circulations such as the positive phase of the WP pattern. The opposite case of minimum $WHD_{NCP}$ in 2010 further supports the mechanism of how EA/WR and WP patterns and associated external factors altered the local climate conditions to impact the $WHD_{NCP}$.

## 1. Introduction

Related to booming economic development, atmospheric pollution has become a serious environmental and social problem in China (Ding et al. 2014; Wang et al. 2016). Particularly after the persistent heavy fog and haze events in January 2013, haze pollution has become more severe (Zhang et al. 2014; Zhao et al. 2014; Li et al. 2015) and has presented certain negative effects on human health (Yin et al. 2011; Chen et al. 2013). The North China Plain (NCP), a location in which the population density is quite high, was one of three haze-prone regions in China. The winter (December–February) haze pollution over NCP (34–43$^{\circ}$N, 114–120$^{\circ}$E) in 2013 and 2014 was the most serious of these events in the past 30 years (Yin et al. 2015a). Therefore, the objective of this study is to examine the related climate conditions (e.g., atmospheric circulation anomalies and external forcings) that were responsible for the extreme haze events in 2014.

It is no doubt that the anthropogenic emissions were the fundamental cause for the long-term variation of haze days (Wang et al. 2013). However, the impact of meteorological conditions is highlighted and the climate conditions are also vital contributors to the interannual variation of haze (Yang et al. 2016; Zhang et al. 2016). For example, the joint effect of fast increase of total energy consumption, rapid decline of Arctic sea ice extent and reduced precipitation and surface winds intensified the haze pollution in central North China after 2000 (Wang et al. 2016). Early studies documented that the East Asian winter monsoon (EAWM) had weakened after 1986, which led to an increase of winter haze days (WHD) over NCP (WHD$_{NCP}$) (Yin et al. 2015a; Yin et al. 2015b; Li et al. 2015). The decline of the preceding autumn (September–November) Arctic Sea Ice (ASI) from 1979 to 2012 greatly intensified haze pollution in eastern China, the variance contribution of which was 45–67% (Wang et al. 2015a). Sea surface temperature (SST) over the subtropical western Pacific (SWP) showed significantly negative correlation with WHD$_{NCP}$. SWP-SST weakened EAWM circulation, leading to a favorable environment for haze with stable atmosphere and potential for hygroscopic growth (Yin et al. 2016a).

The local anti-cyclone anomaly around NCP was the most prominent circulation related to WHD$_{NCP}$ (Yin et al. 2016a; Chen et al. 2015) and was located near the bonding area of two continental Rossby waves, i.e., Eurasia (EU) and East Atlantic/West Russia (EA/WR) patterns. The EU pattern, as defined by Wallace et al. (1981), originated from the high-latitude polar region. The EU positive phase showed negative centers over the polar region (70–80$^{\circ}$N, 60–90$^{\circ}$E) and the

Japan Sea (35–45$^{o}$N, 120–140$^{o}$E), and a positive center over Mongolia and North China (45–55$^{o}$N, 90–110$^{o}$E), which accounted for the severe drought in 2014 (Wang et al. 2015b). Another Eurasian teleconnection, known as the EA/WR pattern (Barnston et al. 1987), was composed of negative centers over Central-North Atlantic and to the north of the Caspian Sea, and positive centers over Europe and North China. The positive phase of these two continental Rossby wave trains might have led to significant warming over the northern portion of Eastern Asia (Liu et al. 2014), indicating weaker cold air. Therefore, we speculated that external forcings such as the SST, ASI and land surface temperature (TS) might impact teleconnection patterns in the atmosphere and then the teleconnection patterns could alter the local climate anomalies to modulate the WHD$_{NCP}$ remotely. Climate research on haze pollution in China is quite a new endeavor but is still insufficient, especially with respect to investigation into the mechanism that causes extreme haze events. Thus, the roles of climate anomalies in the winter haze in 2014 over NCP were investigated in this study and were expected to improve prediction skill for WHD$_{NCP}$.

The remainder of this paper is organized as follows. The data and methods are described in Section 2. The climatic reality of severe WHD$_{NCP}$ in 2014 and associated atmospheric circulations are analyzed in Section 3. Section 4 describes investigation of the physical mechanism using the singular value decomposition (SVD) technique. Brief conclusions and selected discussions are presented in Section 5.

## 2.    Datasets and methods

The China ground observations from 39 NCP stations (Figure 1) collected by the National Meteorological Information Center of China from 1979 to 2015 were used to reconstruct the climatic WHD data, referred to Yin et al. (2016). Due to the quality and temporal range of the data, only four rural stations were qualified and selected (white circles in Figure 1). The routine meteorological measurements included relative humidity, visibility and wind speed that were collected four times per day, i.e., 02:00 local time (LT), 08:00 LT (00:00 UTC), 14:00 LT, and 20:00 LT. In this work, haze is recognized when visibility falls below a certain threshold and the relative humidity is less than 90%. After excluding other weather phenomena that affect visibility, a day with haze occurring at any of the four hours is defined as a haze day. Most of the

visibility observations have switched from manual to automatic since Jan 1$^{st}$, 2014, but the trial stations coded 54511, 54527 and 54623 were switched in 2013. Thus, the visibility threshold is 10 km before Jan 1$^{st}$, 2014 for most stations, but before Dec 1$^{st}$, 2013 for the trial stations. After the switch, the threshold is 7.5 km, according to the China Meteorology Administration's notification (2014). To a certain extent, the WHD data of 2013 and 2014 are more qualitative than quantitative. To avoid continuity problems, the haze data for these two years were processed using composite analysis instead of correlation analysis in this study.

Haze is a multidisciplinary phenomenon that can be represented by visibility and humidity in meteorology, and by the concentration of the atmospheric composition in environmental science. In recent years, the atmospheric compositions involving the concentrations of $SO_2$, $NO$, $NO_2$, $NO_x$, $CO$, $O_3$, $PM_{2.5}$ (by TEOM 1400a) and nephelometric turbidity (NEP) were measured in Shangdianzi and Baolian (an urban site in Beijing, Zhao et al. 2011). The detailed atmospheric composition datasets were collected four times per day from Dec 1$^{st}$, 2014 to Feb 28$^{th}$, 2015. Hourly $PM_{2.5}$ data during 2004—2015 and 2008—2015 at Shangdianzi and Baolian Station respectively, are used in this study. Shangdianzi Station, one of the six regional global atmospheric watch (GAW) stations in China is located at 40 °39'N, 117 °7'E, 293.3 m above sea level. As the only regional background GAW station in North China, the atmospheric composition at this location was chosen to best characterize the natural state or rather the background conditions of the atmosphere (Yao et al. 2012). The $PM_{2.5}$ data were monitored every 5 minutes using two methods: tapered element oscillating microbalance (TEOM) and β-ray (since 2013).

In China, the temporal range and quality of the observed atmospheric compositions cannot support climatic haze research. To demonstrate if the visibility-based reconstruction of haze data is representative, the hourly visibility and concentration of the atmospheric composition were shown, and the correlation coefficients were computed (Figure S1). In addition to significantly positive correlation with $O_3$, the correlation coefficients between visibility and seven compositions were all negative and exceeded the 99.99% confidence level in winter 2014. $PM_{2.5}$ was the main reason for haze pollution, and the correlation coefficients with visibility in Beijing were −0.51 (Baolian) and −0.48 (Shangdianzi). When visibility was less than 7.5 km and 1 km, the mean $PM_{2.5}$ mass concentrations at Baolian were approximately greater than 100 and 200μg/m$^3$,

respectively. Thus, the tendency and magnitude both showed that the derived haze datasets not only agreed with the meteorological standard but also satisfactorily represented the concentration of the atmospheric composition.

Monthly atmospheric data such as wind, geopotential height, temperature and sea level pressure (SLP) are derived from the National Centers for Environmental Prediction/National Center for Atmospheric Research global reanalysis dataset with a horizontal resolution of $2.5\,° \times 2.5\,°$ collected from 1979 to 2015 (Kalnay et al. 1996). The monthly mean planetary boundary layer height (PBLH, $1^o \times 1^o$) was derived from the ERA-Interim dataset (Dee et al. 2011). The monthly mean Extended Reconstructed SST (ERSST) datasets with a horizontal resolution of $2^o \times 2^o$ collected from 1979 to 2015 were obtained from the National Oceanic and Atmospheric Administration (NOAA) (Smith et al. 2008). The ASI extent was calculated using the ASI concentration data from Hadley Center's HadISST1 with a resolution of $1^o \times 1^o$ collected from 1979 to 2015 (Rayner et al. 2003). The EA/WR and WP indices were computed by the NOAA climate prediction center according to the Rotated Principal Component Analysis used by Barnston et al. (1987). The calculation procedure for the EU index was consistent with that of Wang et al. (2015b):

$$\text{EU index} = \left[ -1U\overline{H500}_{(70-80^oN, 60-90^oE)} + 2 \times \overline{H500}_{(45-55^oN, 90-110^oE)} - 1 \times \overline{H500}_{(35-45^oN, 120-140^oE)} \right]/4$$

where H500 represents the geopotential height at 500 hPa and overbars denote the area average.

To verify the covariability between the atmosphere and external forcings, i.e., SST, ASI and TS, SVD and correlation analyses were applied after linear trends were removed. To some extent, the energy consumption varied continuously and linearly in eastern China and the socio-economic components of $WHD_{NCP}$ could be removed primitively by detrending, and then the interannual variability of haze pollution should be mainly the result of climatic anomalies. The anomalies in 2010, 2013 and 2014 were calculated relative to the climatic mean of 1979-2012. The site WHD data were converted into grids using the Cressman interpolation (Cressman, 1959), and $WHD_{NCP}$ was computed as the mean value of the grid data.

## 3.  Variations of $WHD_{NCP}$ and associated atmospheric circulations

According to Figure 2, $WHD_{NCP}$ from 1979 to 2012 can be divided into two decadal stages, i.e., the first stage from 1979 to 1992 with an average of 45.1 days and the second decade from 1993 to 2012 with an average of 34.5 days. It is obvious that

WHD$_{NCP}$ exhibited rapid increase since winter 2010 with a large trend of 7.36 d/yr. NCP is a haze-prone area in which WHD is distributed nonuniformly (Figure 3a). Two regions exhibited greater WHD: the plain to the east of Taihang Mountains and to the south of Yan Mountains (PETSY) and the south of Shandong Province. As shown in Figure 1, there were four rural stations, three of which were located near the Yan Mountains and were corresponding to less WHD. Another rural site near the boundaries of Shandong and Henan (BSH) showed less WHD. Figure 3b shows the WHD anomalies in 2014 with respect to 1979–2012. In addition to a few sites, a larger number of WHD occurred, especially on the BSH (rural area) and the northeast of Hebei. It is notable that WHDs in these two regions show significant increases, filling up the climatic WHD valley as shown in Figure 3a. As a result, the haze-prone area joined together, indicating that the haze pollution was more serious in this region. Actually, the fast increase of WHD in rural area was an obvious reflection of the severe haze disaster in recent years. At the same time, a larger number of WHD occurred on PETSY and the south of Shandong Province. Recently, the haze pollution has become increasingly serious, as shown by the number of haze days and its coverage. The percentages of sites with greater than 30 and 60 hazy days were 71.8% and 51.3%, respectively, in 2014. As shown in Figure S1, the mean PM$_{2.5}$ mass concentrations in Beijing were above 100μg/m$^3$ on hazy days, which indicated serious pollution in 2014. Although data continuity was influenced by the switch of the observation method in 2013, the observation that WHD$_{NCP}$ in 2014 was greater than before showed robustness. Thus, it was reasonable to treat the year 2014 as a typical case for haze pollution over this region. As shown in Figure 2, NCP experienced the least WHD in 2010, which could be analyzed as an inverse case of the extreme haze phenomena.

In the lower and middle layers of the troposphere, the correlation fields between WHD$_{NCP}$ and H500 (UV850) represented obvious EA/WR and WP patterns (Figure 4a–b). The EA/WR pattern originated from the north-central Atlantic and propagated through Europe, the north of Caspian Sea and North China (Barnston et al. 1987; Liu et al. 2014). The WP pattern showed two activity centers, i.e., the broad area of Southeast Asia and the northwest Pacific and the Kamchatka Peninsula (Barnston et al. 1987). In addition to EA/WR, the EU pattern was another continental Rossby wave train over Eurasia, which significantly impacted the climate of East Asia (Liu et al. 2014). Although the EU pattern was unclear in Figure 4a–b, it can be recognized distinctively from the anomalous circulations in 2014 (Figure 4c). The correlation

coefficients between $WHD_{NCP}$ and these three pattern indices were calculated (Table 1). After detrending, $WHD_{NCP}$ showed significant positive correlation with both the EA/WR and WP patterns, indicating the remote impact on $WHD_{NCP}$ from land and sea. The correlation between EU and $WHD_{NCP}$ was initially significantly positive but became insignificant after detrending. Considering that EU could be recognized from low, middle and high layers in winter 2014, the EU pattern was still treated as a possible circulation correlated with $WHD_{NCP}$. These teleconnection patterns might contribute to $WHD_{NCP}$ by impacting the pivotal and local anti-cyclone anomalies (i.e., the local climate conditions) over NCP (Figure 4a—b). The deep and broad positive anomalies resulted in a thinner boundary layer by suppressing vertical movement and progressed easterly to weaken the East Asia Jet Stream (EAJS) indicating weaker meridional cold air. Near surface, the negative SLP anomalies in the Siberia region and the Chinese Mainland and the positive SLP anomalies over the West Pacific led to a reduced pressure gradient and weakened EAWM (Figure 5a). The weaker EAWM induced the southeasterly anomalies and reduced the surface wind speed (Figure 5b), indicating weaker cold air and warmer land surface over NCP (Figure 5a). Influenced by the circulations with weaker cold air, horizontal diffusion of the atmospheric particulates was impeded. Near NCP, the anomalous anti-cyclone also existed, indicating weaker vertical motion. The main physical process was that the climate teleconnection patterns altered the local climate conditions, and then influenced the dispersion capacity the local atmosphere. That is, when the positive pattern of EA/WR, WP and EU occurred together or partly, the anomalous anti-cyclone over NCP and Japan Sea was enhanced from surface to the middle troposphere, thus, the convection or vertical motion was confined. The southerly anomalies on the left side of this anti-cyclone weakened the cold air and wind speed, but brought about humid flow. Under such local climate anomalies, the vertical and horizontal diffusion of atmospheric particulates were both restricted and then the pollutant gathered in a narrow space and resulted in the occurrence of haze.

In winter 2014, many extreme climatic events occurred, such as severe drought and high temperatures in NCP (Wang et al. 2015). Corresponding external forcings should be observed as the background of these extreme synoptic and climatic events, which might persistently impacted the winter atmospheric circulations and led to an irregular distribution of teleconnection patterns in winter. It should be noted that EA/WR pattern in winter 2014 was distributed slightly westwards and broadly. Nevertheless, the three eastern centers of EA/WR, WP and EU patterns could be recognized (Figure 4c). The linkage

anti-cyclone of these three teleconnection patterns was enhanced and modulated the local climate conditions. The NCP area was influenced by the anomalous high resulting in lower PBLH (Figure 6). The southerly at the high latitudes deadened the cold air from its main source and enhanced accumulation of local air pollutants, so the atmospheric matters gathered easily. Near surface, the negative SLP anomalies occupied the whole Asian continent and Japan Sea that weakened the continental cold high and stimulated significant southerly anomalies to the north of $50^{o}$N with the weaker Aleutian low. The weaker EAWM circulations near surface blocked the cold air from high latitudes and resulted in warmer surface (Wang et al. 2015). There were positive SLP anomalies over South China Sea and East China Sea that induced southerly and smaller surface wind speed over the coastal area in the east of China (Figure 5c-d). The dispersion of atmosphere over the NCP was limited, so the high pollutant emissions were concentrated in a narrow space and severe haze events occurred easily. Large scale circulations, such as the negative phase of EA/WR and WP, were quite clear in 2010 (Figure 4d). Near surface, the anomalous circulations were distributed similarly but almost opposite, i.e., the stronger continental cold high and oceanic low (Figure 5e), the northerly and stronger surface wind over NCP (Figure 5f), and the higher PBLH (Figure 6). The atmospheric diffusivity was heightened by the stronger cold air and vertical movement. This observation further supports the speculation that the anomalous EA/WR and WP patterns contributed to significant changes in $WHD_{NCP}$ by altering the local climate conditions. The anomalies of the EU pattern in winter 2010 were not as significant as those in 2014. As shown by Table 1, the relationship between $WHD_{NCP}$ and the EU index weakened after detrending, illustrating that the correlation was much weaker than the other two patterns.

## 4. Possible mechanisms for the winter haze in 2014

In the above discussion, we addressed the associated circulations that were characterized by EA/WR, WP and EU patterns and that contributed to the extreme haze pollution in 2014. Wang et al. (2015a) found that the pre-autumn ASI could significantly impact the WHD in the east of China. As another efficient external forcing, the negative SWP-SST anomalies (SSTA) markedly intensified $WHD_{NCP}$ (Yin et al. 2016a). Thus, the question arises as to whether these factors could have caused the extreme haze pollution in 2014.

Since 1979, the pre-autumn ASI declined substantially, which might impact the winter climate of East Asia (Liu et al. 2012). Furthermore, this decreasing trend of the ASI intensified since 2006 (Wang et al. 2015a). The relationship between the pre-autumn ASI and the circulations was analyzed using the SVD method, explaining 51.6% and 17.9% of the variance by the first (SVD1) and second (SVD2) mode, respectively. The correlation coefficient of the first temporal series was 0.76, which was significantly above the 99% confidence level. The excited anti-cyclonic (A) or cyclonic (C) activity centers of H500 in winter were located over the north Atlantic, Europe, to the north of Caspian Sea and around Mongolia and North China, which appeared as the positive phase of the EA/WR pattern (Figure 7a). The corresponding pre-autumn ASI, with more sea ice (SI) over the Barents Sea, Kara Sea and Laptev Sea (BKLSI) and less SI in the East Siberian Sea (ESSI) could have contributed to the positive EA/WR pattern (Figure 7b). The anomalous ASI of the Eastern Hemisphere in 2014 was similar to that shown in Figure 7b. According to the SVD results, the pre-autumn ASI anomalies in 2014 might have triggered the EA/WR pattern in the atmosphere, especially the anomalous high over NCP, and led to greater $WHD_{NCP}$. In autumn 2010, less BKLSI and more ESSI (Figure S2) occurred, which might have induced less WHD over NCP.

In winter 2014, the TS anomalies of the broad region of Eurasia (40–70$^{o}$N) were obviously positive. EA/WR is a famous continental teleconnection, and a significant land-air interaction could influence the EA/WR pattern. To illustrate this concept, an SVD analysis was performed between the detrended and normalized winter TS and H500 (Figure 8). The explained variance and correlation coefficient of the first component were 53.3% and 0.73, respectively. The spatial coefficients of TS were distributed similarly with TS anomalies in 2014. The associated circulations displayed three anomalous centers, i.e., a positive center over Europe, negative anomalies to the north of Caspian Sea, and another positive center over Mongolia and North China, which were also the three atmospherically active centers of EA/WR on the continent that occurred in winter 2014 (Figure 5c). These results indicated that the TS anomalies of winter 2014 could have induced or intensified the positive EA/WR phase and resulted in the extreme $WHD_{NCP}$ case in 2014. In winter 2010, the cooler land surface of mid-latitude Eurasia (Figure S3) could have contributed to the negative EA/WR phase. The diagnostic analyses of two inversely extreme $WHD_{NCP}$ cases enhanced the relationship between the winter mid-latitude Eurasia TS and $WHD_{NCP}$.

Figures 7 and 8 also show negative anomalies over the polar area (70–80$^{o}$N, 60–90$^{o}$E) and a positive center over Mongolia

and North China (45–55$^o$N, 90–110$^o$E), which can be recognized as the positive EU phase. When BKLSI was above normal and ESSI was below normal or the surface of mid-latitude Eurasia was warmer, these two centers could be stimulated or enhanced, and represented favorable circulations for haze occurrence. Prior to this study, Yin et al. (2016) found that the pre-autumn SSTA of the Pacific excited the anti-cyclone anomalies over North China and resulted in more WHD$_{NCP}$. The SST in October and November (ON) and winter H500 were decomposed using the SVD technique to reveal the main relationships. The explained variance and the correlation coefficient of the first component were 60.2% and 0.73, respectively. An EU-like pattern occurred in the first mode of H500, i.e., cyclonic anomalies over the polar area and anti-cyclonic anomalies over Mongolia and North China (Figure 9a). The associated SST was cooler over the northwest Pacific, involving Kuroshio and its extension, and warmer over the central-east Pacific and Alaska Gulf (Figure 9b). Another atmospheric response was the weaker East Asia trough, indicating weaker EAWM and cold air. Figure 9c shows the ON SSTA of the Pacific in 2014, which appeared to be similar to the SST SVD1 distribution and stimulated haze-prone responses. In 2010, the ON SSTA in the Pacific did not show a well-organized opposite pattern (Figure S4), but a cooler SST was observed on the central-east Pacific and Alaska Gulf, which were of benefit to haze occurrence.

Both EA/WR and EU were continental teleconnection patterns that propagated over Eurasia. In contrast, the WP pattern was located over the junction of the marine and mainland areas, and its positive phases were composed of a broad positive center over the northwest Pacific and Southeast Asia and a negative center over Kamchatka Peninsula (Barnston et al. 1987). From Table 1, the stronger WP positive phase could have contributed to more WHD$_{NCP}$ and could also have been partially responsible for the severe haze event in 2014. To identify the causes of such circulations, we performed an SVD analysis between winter H500 and pre-autumn TS, with explained variance of 39.8% and 20.7%, respectively, for SVD1 and SVD2. The results of SVD2 showed significant anomalies in the two centers of the positive WP pattern and anomalous TS distributed from the southwest to the northeast in Eurasia (Figure 10). The correlation coefficient of the SVD2 time series was 0.62, exceeding the 99% confidence level. Similar to the SVD2 representation, two "southwest to northeast" anomalous TS distributions also occurred in autumn 2014, i.e., negative anomalies from the Caspian Sea to Baikal Lake and a positive anomalous belt from Southwest China to Northeast China (Figure 10c). Referring to SVD2, these "southwest to northeast"

TS anomalies could have induced atmospheric responses over the junction of the Pacific and Asian land, which resembled the WP pattern and impacted the local circulations over the NCP area. In autumn 2010, the positive belt of TS from the Caspian Sea to Baikal Lake was significant, and the land surface of the Tibet Plateau was warmer than normal (Figure S5).

These two anomalies were dramatically inverted with the SVD2 results and could have stimulated a negative WP pattern. As argued by two opposed cases, the mechanism for how the EA/WR and WP patterns and associated external factors impacted the $WHD_{NCP}$ were confirmed.

## 5.   Conclusions and discussions

Except for a few sites, haze pollution over NCP in winter 2014 was the severest in the past 30 years. On the BSH area and

245 the northeast of Hebei, WHD increased significantly and invaded into the rural area, illustrating a joint and broad severe haze-prone region. The $PM_{2.5}$ concentration at a GAW station and an urban site were almost equal in winter 2010 and 2014, so there was no evidence that the emissions were more in 2014 than 2010. The climate anomalies played key roles in the formation of heavy haze pollution case in 2014. In the lower and middle troposphere, the positive phases of EA/WR, WP and EU patterns modulated the local anti-cyclone anomalies over North China. The anti-cyclone anomalies over East Asia

not only resulted in stable atmospheric stratification and a thinner boundary layer but also led to a southeasterly anomaly that weakened the cold air but enhanced the moisture transport. The atmospheric matters would accumulate easily both on the vertical and horizontal direction. In winter 2014, the teleconnection patterns, such as EA/WR, EU and WP, combined to alter the local climate conditions to contribute to the extreme haze case. SVD analyses indicated that the pre-autumn ASI anomalies of the Eastern Hemisphere and the warmer winter surface of Eurasia could have induced or intensified the

responses in the atmosphere, resembling a positive EA/WR pattern. These two external forcings, together with the SSTA in Pacific (i.e., cooler in the northwest Pacific and warmer in the central-east Pacific and Alaska Gulf) might stimulate or enhance positive EU-like patterns. In autumn 2014, the "southwest to northeast" anomalous TS belts were other factors that efficiently intensified the haze pollutions, which resulted in a positive phase of the WP pattern. The case of 2010, with the least $WHD_{NCP}$, was diagnosed as an opposite case, which further supports the speculation that the anomalous EA/WR and

WR patterns and associated external forcings have a significant impact on $WHD_{NCP}$. Additionally, as pointed out by Wang et al. (2015), the Asian high temperature and drought from summer in 2014 was an extreme climate event. Our studies proved that this previous extreme climate event possibly contributed to the serious haze event in the following winter.

The rapid increase of $WHD_{NCP}$ began in January 2013, and in winter 2013 and 2014, $WHD_{NCP}$ was significantly greater than before. The number of $WHD_{NCP}$ in 2013 and 2014 was greater than 50 and almost equal to each other in the two years (Figure 2). Therefore, the causes of serious haze in 2013 should also be discussed. The EA/WR pattern was well organized and showed a positive phase that was distributed slightly eastwards (Figure S6). Influenced by the upper EA/WR pattern, the surface wind speed was slower and the PBLH was lower, illustrating the horizontal and vertical dispersion of the atmosphere was weaker (Figure S7). The EU and WP patterns were unclear. The source region of EU even showed characteristics of a negative phase. According to the 2010 and 2014 case studies, the proceeding and simultaneous external forcings could have impacted the $WHD_{NCP}$. In contrast, the pre-autumn ASI and TS and winter TS in 2013 did not show features similar to those in 2014 (Figure S8). The pre-autumn Pacific SSTA, which was slightly negative in the northwest Pacific and positive in the Alaska Gulf and central-east Pacific, could have stimulated positive anomalies over NCP and weakened the East Asia trough. Thus, it can be observed that, among many external reasons for the extreme haze in 2014, only the pre-autumn Pacific SSTA was distributed similarly in 2013. The EA/WR Rossby wave train was the prominent circulation contributing to $WHD_{NCP}$ in 2013, with a source located over the central-north Atlantic. We speculate that the air-sea interaction over north Atlantic excited the EA/WR pattern in the atmosphere and influenced $WHD_{NCP}$ remotely. The correlation coefficients between the EA/WR index and the pre-autumn SST in Atlantic were calculated and were significantly positive to the south of Greenland (Figure S9b). When the SSTA to the south of Greenland was positive, the responses similar to EA/WR occurred in the atmosphere. The pre-autumn SSTA in the Atlantic in 2013 was similar to that shown in Figure S9b and might have remotely impacted $WHD_{NCP}$ via the EA/WR pattern. It should be noted that the SSTA of the key region in the Atlantic was negative and had an adverse effect on $WHD_{NCP}$ in 2014.

From the point of facilitating a larger amount of $WHD_{NCP}$, the associated circulations and external forcings in 2013 were different from that in 2014, but the serious situations of haze were almost the same. In our study, we assumed that the energy

consumption linearly increased in the recent years. On such hypothesis, the human activities mainly impacted the long-term trend of $WHD_{NCP}$. After removal of the linear trend, the interannual variability of haze pollution should be mainly the result of climatic anomalies. The Shangdianzi site is the only GAW station in North China and was chosen to reflect the natural or background situation of the atmosphere. The mean mass concentrations of $PM_{2.5}$ in winter from 2004 to 2014 are plotted in Figure 11. The concentration in winter 2013 increased abruptly up to nearly twice that in 2010 and 2014 and was the highest in the observation history that broke down our assumption. Furthermore, the $PM_{2.5}$ concentration of an urban site, Blaolian station, was also much higher than 2010 and 2014. Even the anomalous circulations were not benefit enough for haze occurrence, the joint effect of highest pollution emissions and climate conditions could result in the serious haze event. Documented by these three case studies, the influences of the highest $PM_{2.5}$ were the fundamental cause, and the associated atmospheric anomalies and external forcings played key roles in the severe haze pollution. In the case studies, 2010 and 2014 exhibited approximately equal $PM_{2.5}$ concentrations of the background atmosphere, but the associated circulations and external forcings were slightly different. It is possible that not all of the above factors might be found in a specific case study, i.e., a few of these factors played the essential roles and led to the characteristics of that case (See a brief summary of the impacts of these factors on $WHD_{NCP}$ in Table 2). In this study, we focused on the roles of climate anomalies and the impact of human activities will be studied in our future work. To separate the contributions quantitatively by numerical models or advanced statistical approaches would be a meaningful task that was helpful to the interpretation of mechanism and the seasonal prediction (Yin et al. 2016b).

**Acknowledgements**

This research was supported by the National Natural Science Foundation of China (Grants: 41421004 and 41210007) and CAS-PKU Partnership Program.

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

**Tables and Figures captions:**

**Table 1.** Correlation coefficients between $WHD_{NCP}$ and three teleconnection (EA/WR, WP and EU) indices. A single asterisk (*) indicates that the result exceeded the 95% confidence level, and double asterisks (**) indicate that it exceeded the 99% confidence level. Note: OS means 'original sequence', and 'detrended' means that the linear trend was removed.

**Table 2.** Summary of the various influnce factors for $WHD_{NCP}$. The "+++" indicates "more important"; "++" indicates "important", "+" indicates "less important" , and blank indicates "not important".

**Figure 1.** Topographic map (shading; unit: m) of North China and the locations of 39 NCP observation sites (Urban: black circle, Rural: white circle). The NCP area is represented by a black rectangle, and the names of provinces and mountains are written in red and white, respectively.

**Figure 2.** Variation of $WHD_{NCP}$ from 1979–2014 (Units: days), the error bar represents one standard error among the measured sites. For 2013–2014, the thresholds of 7.5 km (blue) and 10 km (gray) are both shown, and the dashed lines indicate the mean values for 1979–1992 and 1993-2012, respectively.

**Figure 3.** Distributions of WHD from 1979 to 2012 (a) and anomalies in 2014 relative to 1979-2012 (b).

**Figure 4.** Correlation coefficients between $WHD_{NCP}$ and winter circulations from 1979 to 2012 with linear trend was removed (a, b), and circulation anomalies in 2014 (c, d) and 2010 (e, f). The circulations in (a, c, e) are TS (shade) and SLP (contour) and those in (c, d, f) are surface wind speed (shade) and wind vector (arrow).

**Figure 5.** Correlation coefficients between $WHD_{NCP}$ and winter H500 (a) / UV850 (b) from 1979 to 2012. The linear trend was removed, and shade indicates that CC exceeds the 95% confidence level. UV850 (arrow) and speed (shade) anomalies in winter 2014 (c) and 2010 (d). A and C represent anti-cyclone and cyclone, respectively.

**Figure 6.** The difference of averaged wintertime PBLH between 2014 and 2010

**Figure 7.** Heterogeneous correlation map of the first SVD model for detrended and normalized (a) H500 during DJF and (b) ASI during SON 1979–2014; (c) ASI anomaly during SON 2014. A and C represent anti-cyclone and cyclone, respectively.

**Figure 8.** Heterogeneous correlation map of the first SVD model for detrended and normalized (a) H500 during DJF and (b) TS during DJF 1979–2014; (c) TS anomaly during DJF 2014. A and C represent anti-cyclone and cyclone, respectively.

**Figure 9.** Heterogeneous correlation map of the first SVD model for detrended and normalized (a) H500 during DJF and (b) SST during ON 1979–2014; (c) SST anomaly during ON 2014. A and C represent anti-cyclone and cyclone, respectively.

**Figure 10.** Heterogeneous correlation map of the second SVD model for detrended and normalized (a) H500 during DJF and (b) TS during SON 1979–2014; (c) TS anomaly during SON 2014. A and C represent anti-cyclone and cyclone, respectively.

**Figure 11.** Mean of $PM_{2.5}$ concentration in winter at Shangdianzi Station from 2004 to 2014 as measured by the TOEM (solid) and β-ray (dash) method. The error bar represents one standard error among the different measured hours.

**Figure S1.** Visibility of Beijing (green) and atmospheric compositions at BaoLian (blue) and Shangdianzi (red) stations at 02:00, 08:00, 14:00 and 20:00 LT from 1[st] Dec 2014 to 28[th] Feb 2015. The eight compositions included here are $SO_2$, NO, $NO_2$, $NO_x$, CO, $O_3$, $PM_{2.5}$ and NEP from top to bottom. The correlation coefficient was recorded as "CC", and the "N" denotes the number of composition samples. The total number of visibility observations was 360, which was adjusted to match the "N" of each composition after quality control and to compute CC.

**Figure S2.** Anomalies of pre-autumn ASI in 2010

**Figure S3.** Anomalies of winter TS in 2010

**Figure S4.** Anomalies of ON SST in 2010

**Figure S5.** Anomalies of pre-autumn TS in 2010

**Figure S6.** UV850 (arrow) and speed (shade) anomalies in 2013. A and C represent anti-cyclone and cyclone, respectively.

**Figure S7.** Anomalies of surface wind speed (contour) and PBLH (shade) in winter 2013

**Figure S8.** Anomalies of external forcings in 2013. (a) ASI in pre-autumn, (b) TS in winter, (c) SST in Oct and Nov, and (d) TS in pre-autumn.

**Figure S9.** Correlation coefficients between EA/WR index and H500 (a) / Atlantic SST in pre-autumn (b). Pre-autumn SST anomaly during SON in 2013 (c) and 2014 (d)

**Table 1. Correlation coefficients between WHD$_{NCP}$ and three teleconnection (EA/WR, WP and EU) indices. A single asterisk (\*)**
**indicates that the result exceeded the 95% confidence level, and double asterisks (\*\*) indicate that it exceeded the 99% confidence**
**level. Note: OS means 'original sequence', and 'detrended' means that the linear trend was removed.**

| Pattern | OS | Detrended |
|---|---|---|
| EA/WR | 0.36* | 0.43** |
| WP | −0.06 | 0.41* |
| EU | 0.34* | −0.13 |

**Table 2. Summary of the various influnce factors for WHD$_{NCP}$. The "+++" indicates "more important"; "++" indicates**
**"important", "+" indicates "less important" , and blank indicates "not important".**

| Factors | 2010 | 2013 | 2014 |
|---|---|---|---|
| PM$_{2.5}$ concentration | ++ | +++ | ++ |
| Local surface wind speed | ++ | ++ | ++ |
| Local PBLH | ++ | ++ | ++ |
| EA/WR | ++ | ++ | ++ |
| WP | ++ | | ++ |
| EU | | | ++ |
| Pre-autumn ASI | ++ | | ++ |
| Winter TS | ++ | | ++ |
| ON Pacific SSTA | + | ++ | ++ |
| Pre-autumn TS | ++ | | ++ |
| SON Atlantic SSTA | | ++ | |

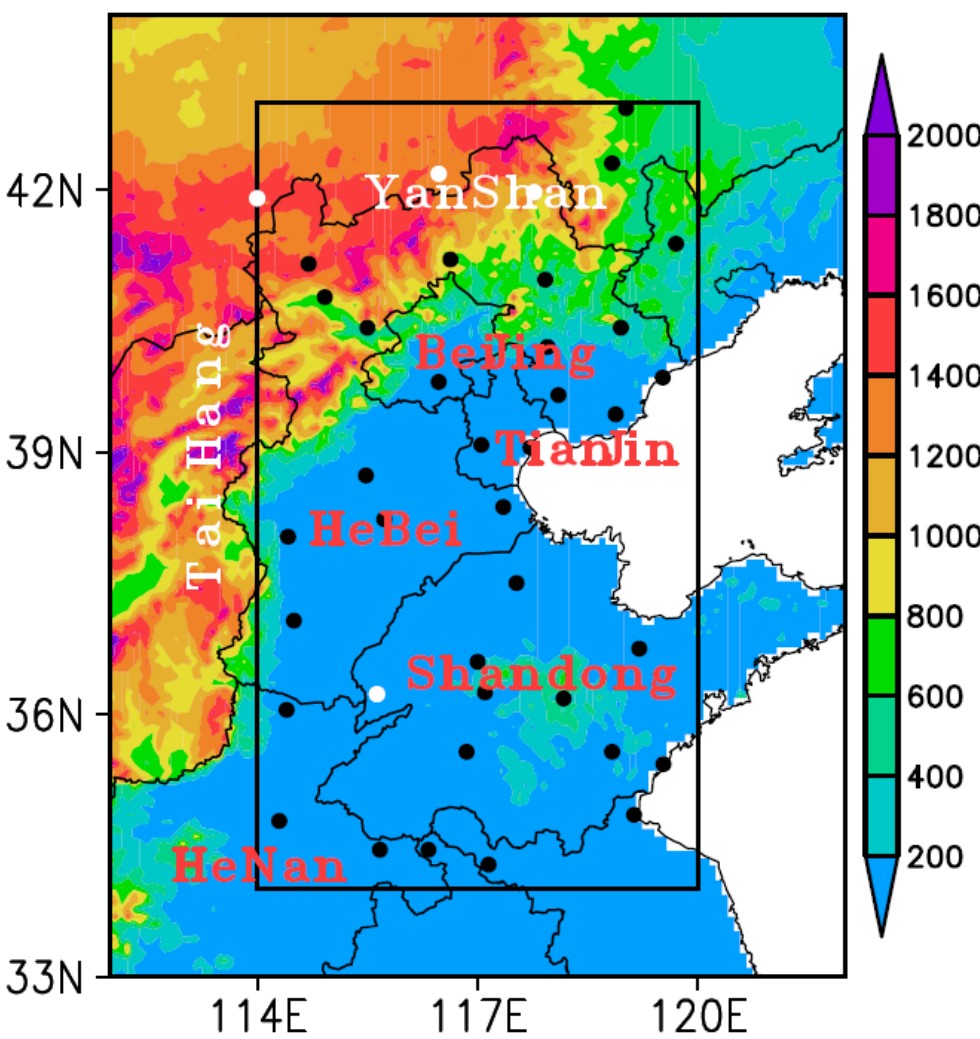

**Figure 1. Topographic map (shading; unit: m) of North China and the locations of 39 NCP observation sites (Urban: black circle, Rural: white circle). The NCP area is represented by a black rectangle, and the names of provinces and mountains are written in red and white, respectively.**

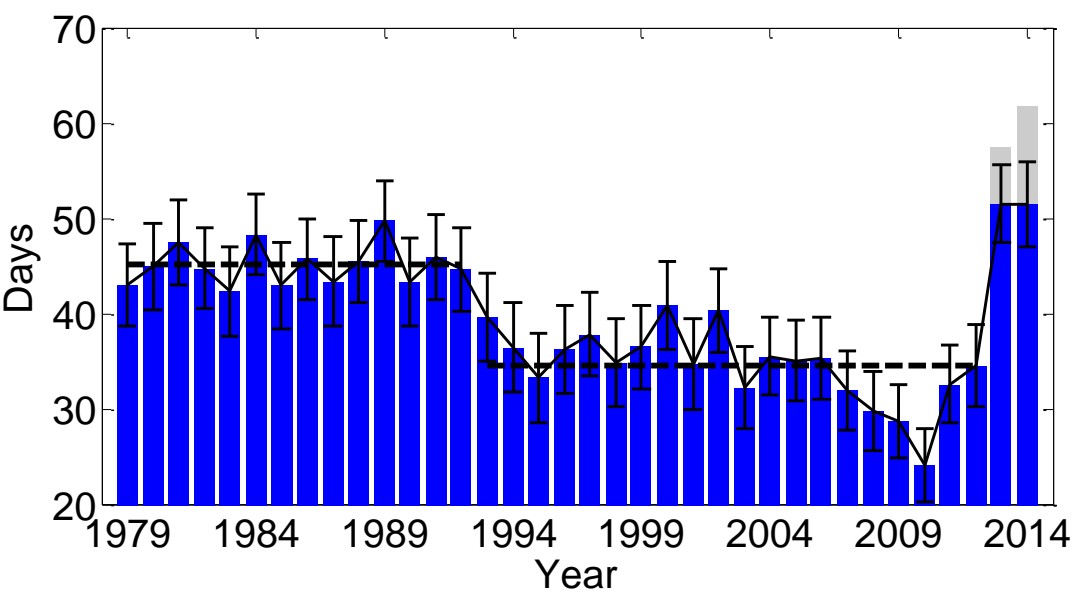

**Figure 2. Variation of WHD$_{NCP}$ from 1979–2014 (Units: days), the error bar represents one standard error among the measured sites. For 2013–2014, the thresholds of 7.5 km (blue) and 10 km (gray) are both shown, and the dashed lines indicate the mean values for 1979–1992 and 1993-2012, respectively.**

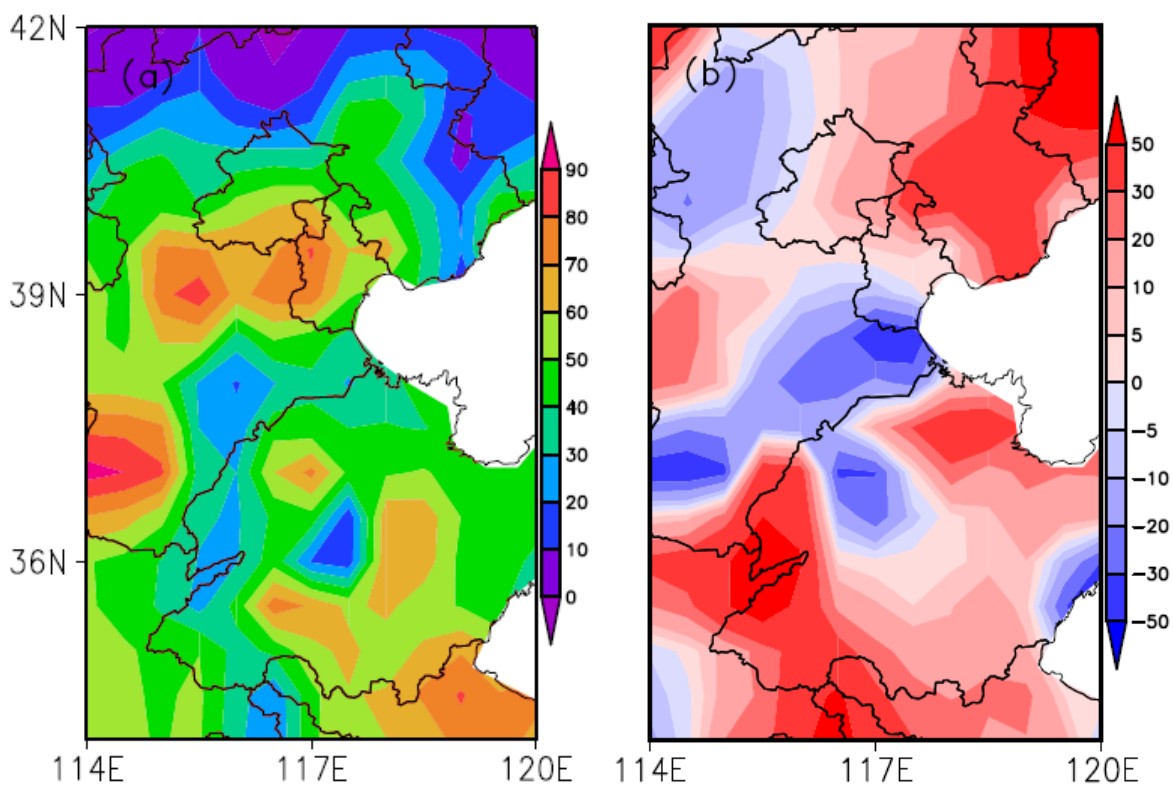

**Figure 3. Distributions of WHD from 1979 to 2012 (a) and anomalies in 2014 relative to 1979-2012 (b).**

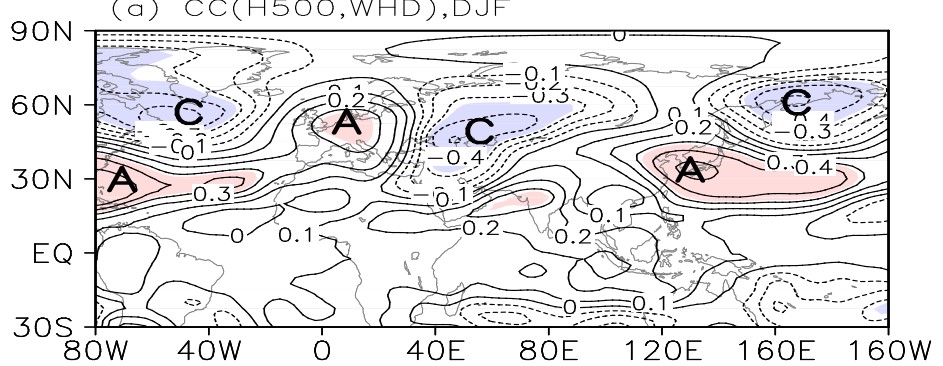

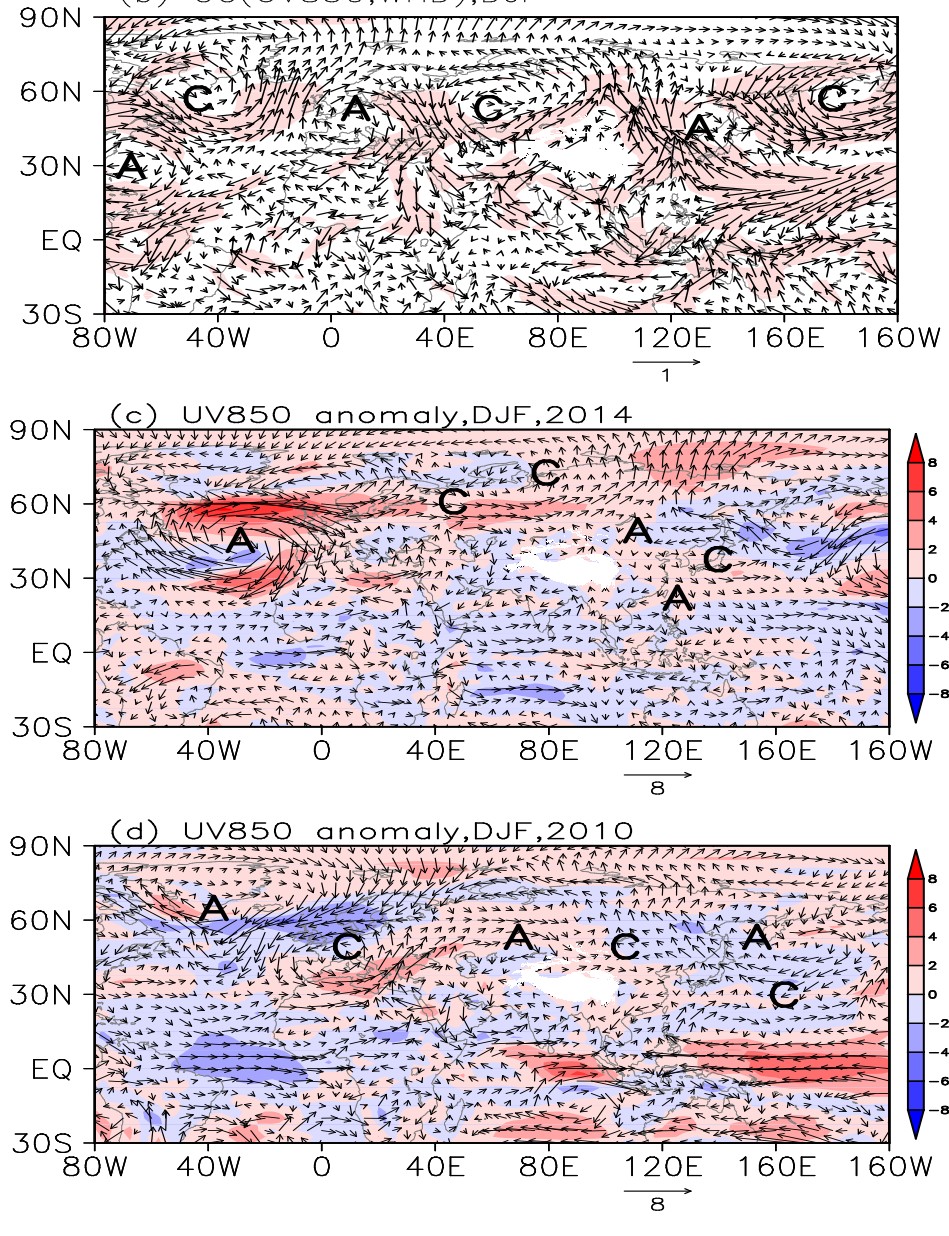

**Figure 4. Correlation coefficients between WHD$_{NCP}$ and winter H500 (a) / UV850 (b) from 1979 to 2012. The linear trend was removed, and shade indicates that CC exceeds the 95% confidence level. UV850 (arrow) and speed (shade) anomalies in winter 2014 (c) and 2010 (d). A and C represent anti-cyclone and cyclone, respectively.**

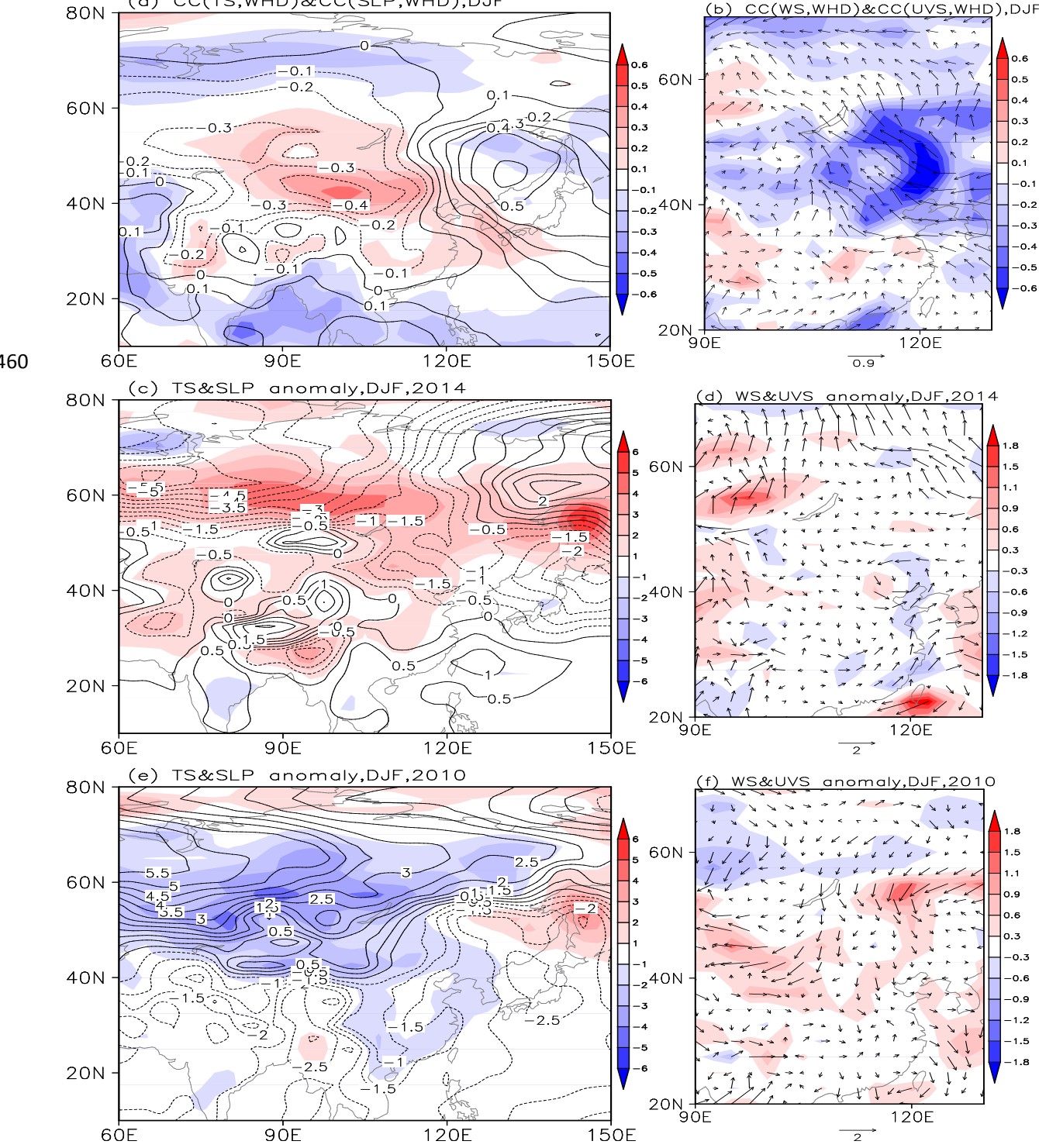

**Figure 5. Correlation coefficients between WHD$_{NCP}$ and winter circulations from 1979 to 2012 with linear trend was removed (a, b), and circulation anomalies in 2014 (c, d) and 2010 (e, f). The circulations in (a, c, e) are TS (shade) and SLP (contour) and those in (c, d, f) are surface wind speed (shade) and wind vector (arrow).**

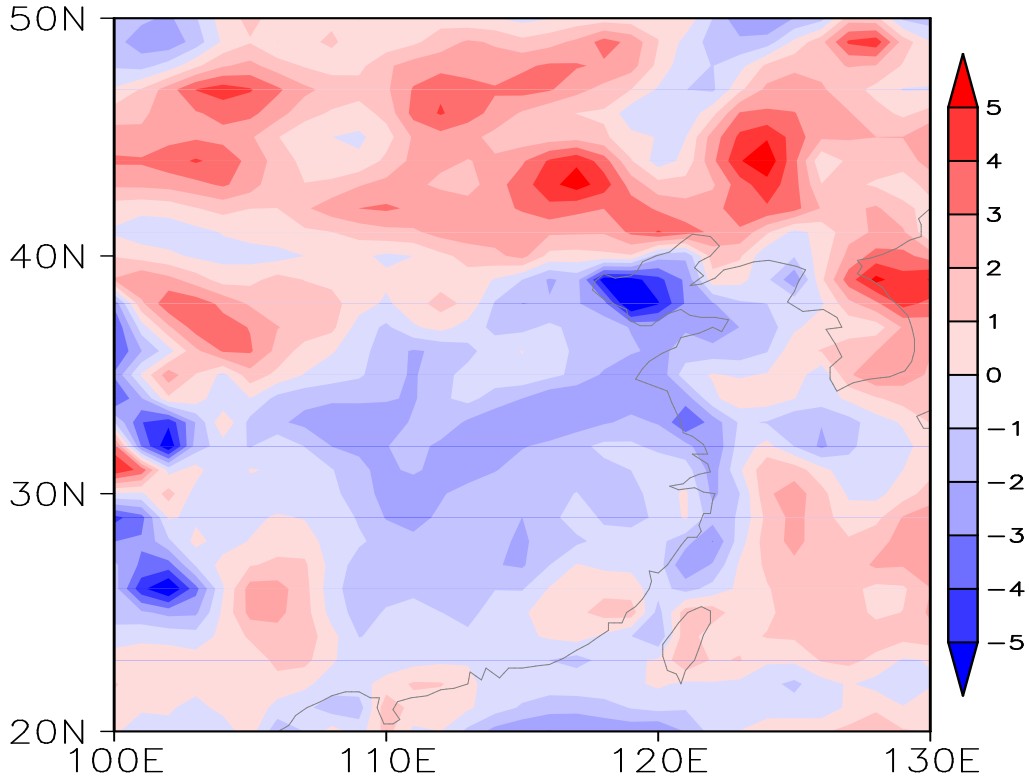

**Figure 6. The difference of averaged wintertime PBLH between 2014 and 2010**

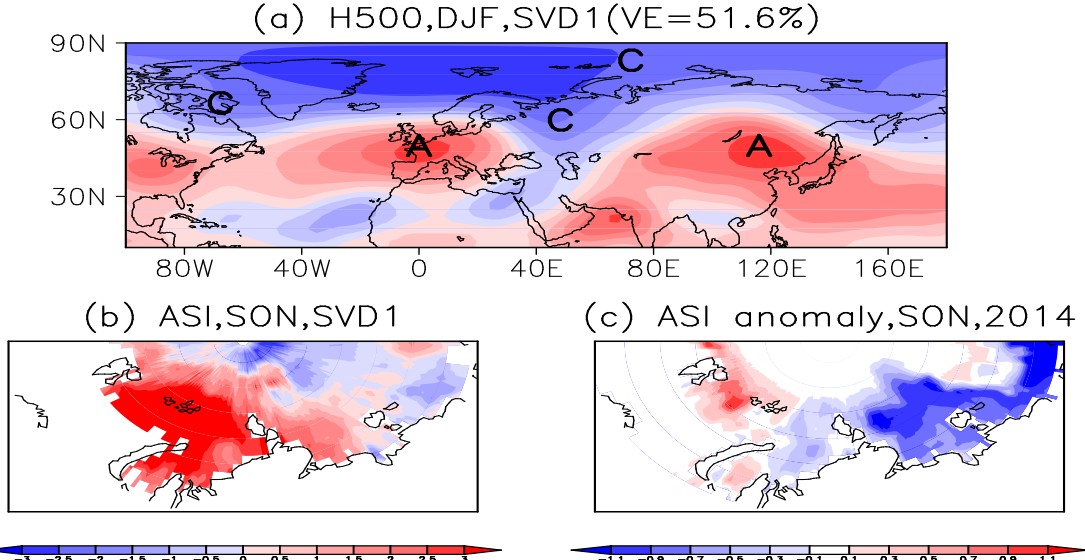

**Figure 7. Heterogeneous correlation map of the first SVD model for detrended and normalized (a) H500 during DJF and (b) ASI during SON 1979–2014; (c) ASI anomaly during SON 2014. A and C represent anti-cyclone and cyclone, respectively.**

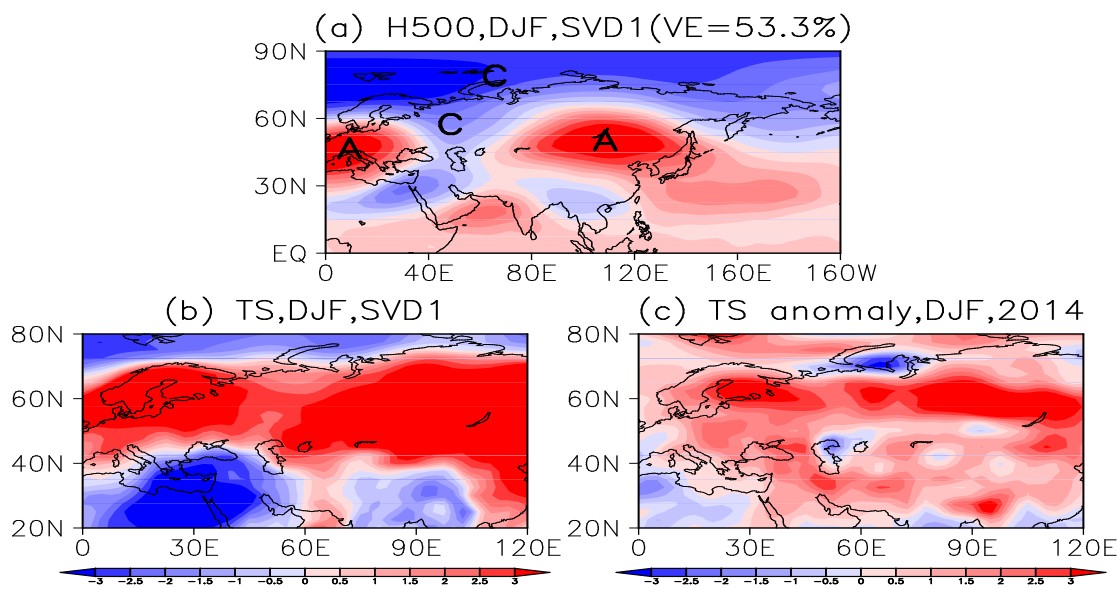

**Figure 8. Heterogeneous correlation map of the first SVD model for detrended and normalized (a) H500 during DJF and (b) TS during DJF 1979–2014; (c) TS anomaly during DJF 2014. A and C represent anti-cyclone and cyclone, respectively.**

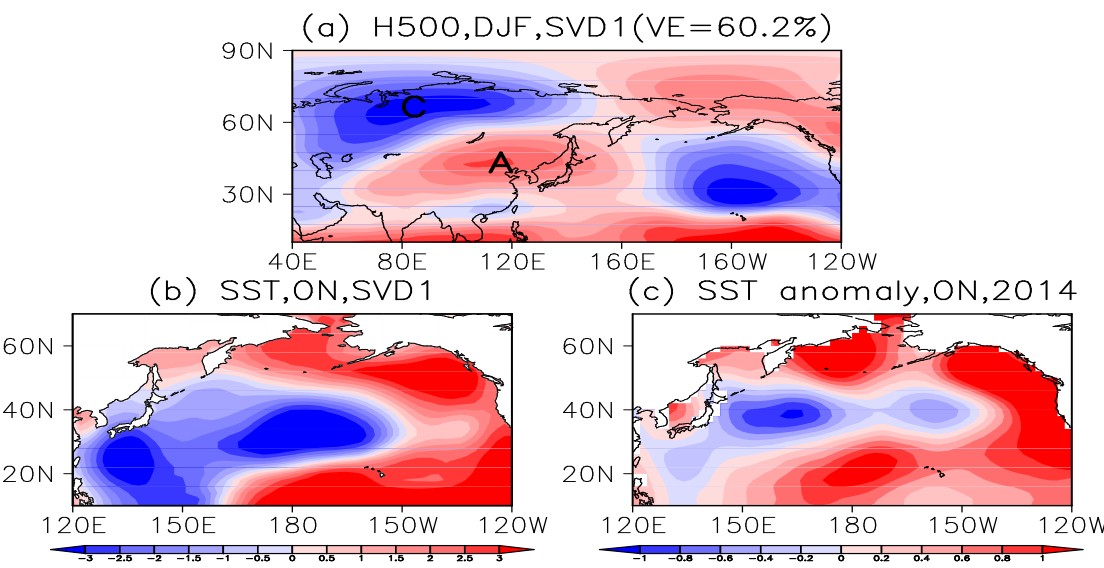

**Figure 9. Heterogeneous correlation map of the first SVD model for detrended and normalized (a) H500 during DJF and (b) SST during ON 1979–2014; (c) SST anomaly during ON 2014. A and C represent anti-cyclone and cyclone, respectively.**

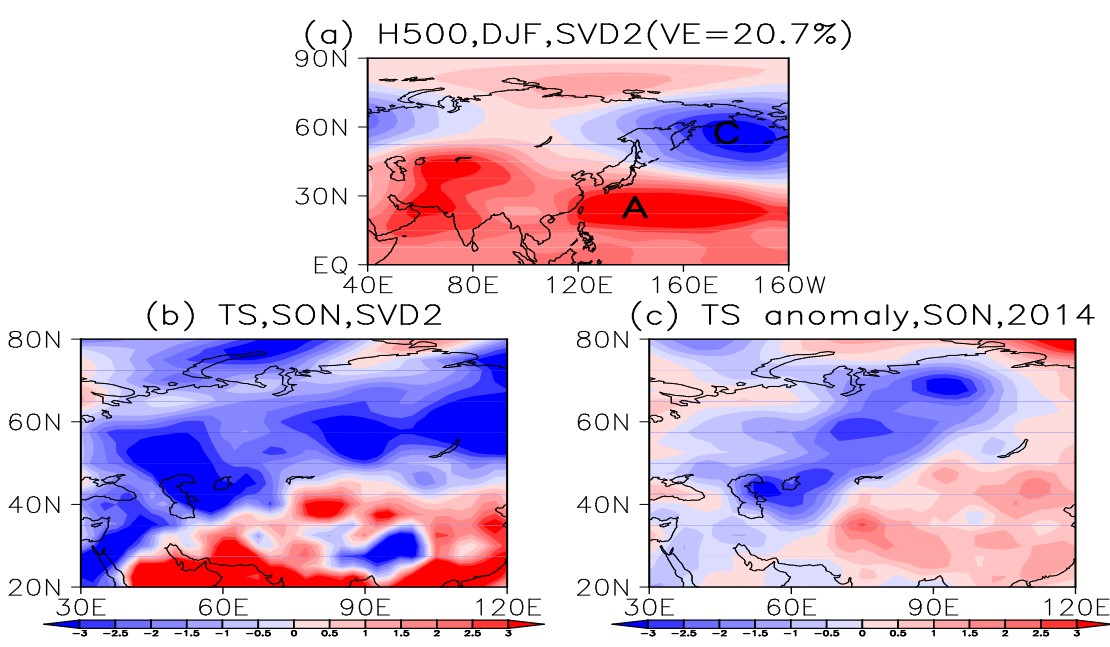

**Figure 10. Heterogeneous correlation map of the second SVD model for detrended and normalized (a) H500 during DJF and (b) TS during SON 1979–2014; (c) TS anomaly during SON 2014. A and C represent anti-cyclone and cyclone, respectively.**

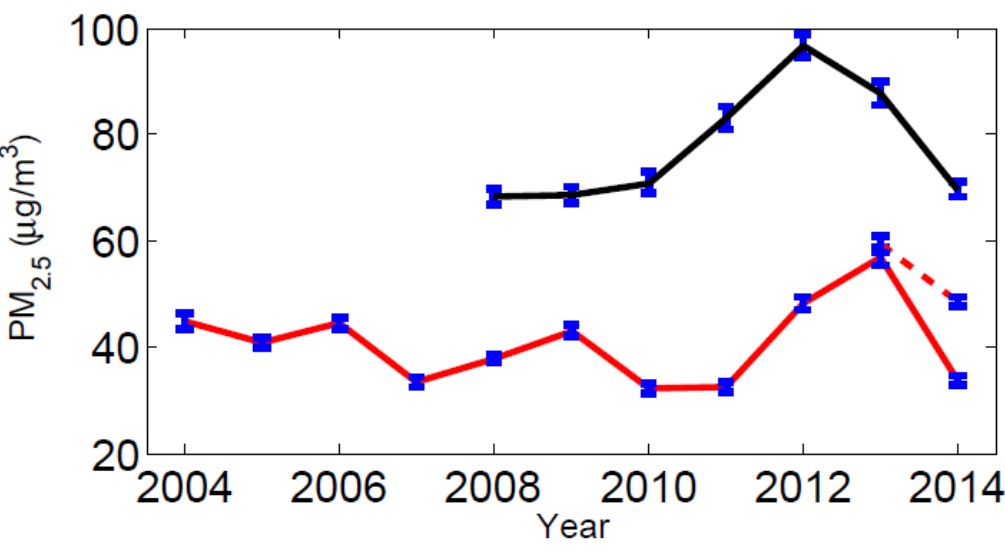

**Figure 11. Mean PM$_{2.5}$ concentration in winter at Shangdianzi (red; measured by the TOEM (solid) and β-ray (red dash) method) and Baolian (black) Station. The error bar represents one standard error among the different measured hours.**

**Supplemental Materials**

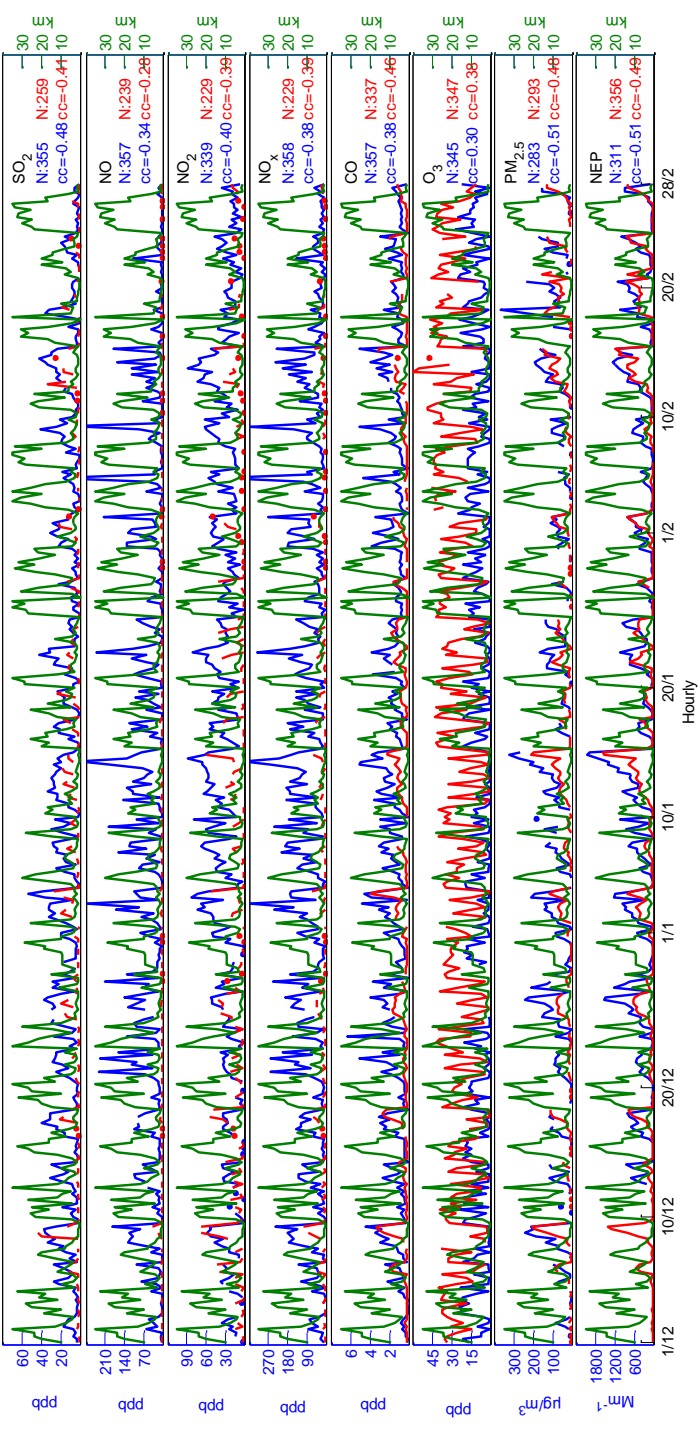

**Figure S1.  Visibility of Beijing (green) and atmospheric compositions at BaoLian (blue) and Shangdianzi (red) stations at 02:00, 08:00, 14:00 and 20:00 LT from 1$^{st}$ Dec 2014 to 28$^{th}$ Feb 2015. The eight compositions included here are $SO_2$, NO, $NO_2$, $NO_x$, CO, $O_3$, $PM_{2.5}$ and NEP from top to bottom. The correlation coefficient was recorded as "CC", and the "N" denotes the number of composition samples. The total number of visibility observations was 360, which was adjusted to match the "N" of each composition after quality control and to compute CC.**

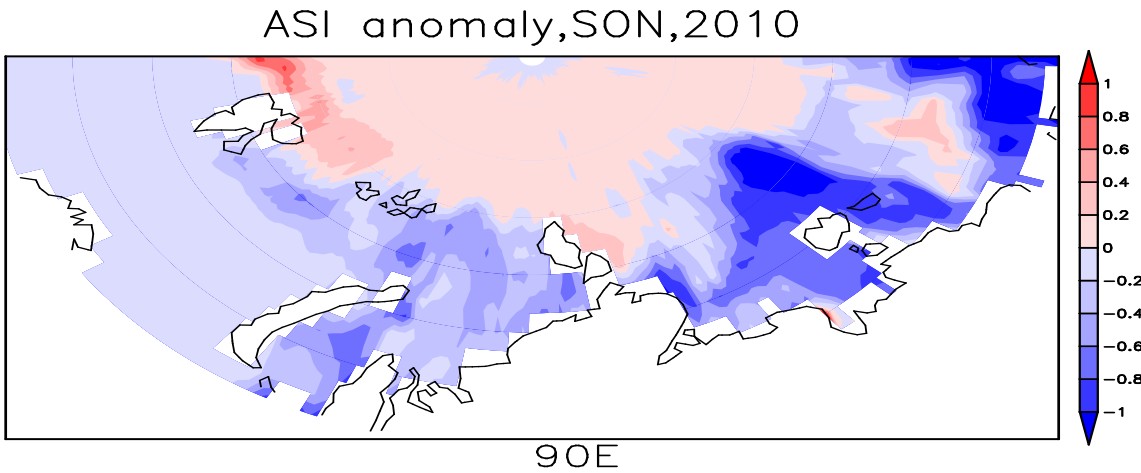

**Figure S2. Anomalies of pre-autumn ASI in 2010**

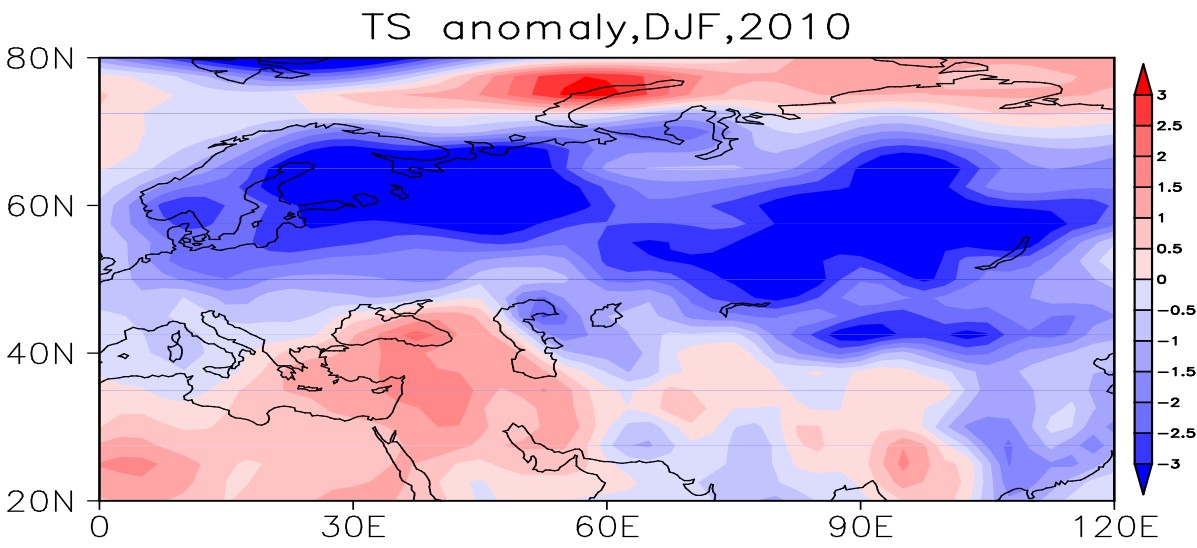

**Figure S3. Anomalies of winter TS in 2010**

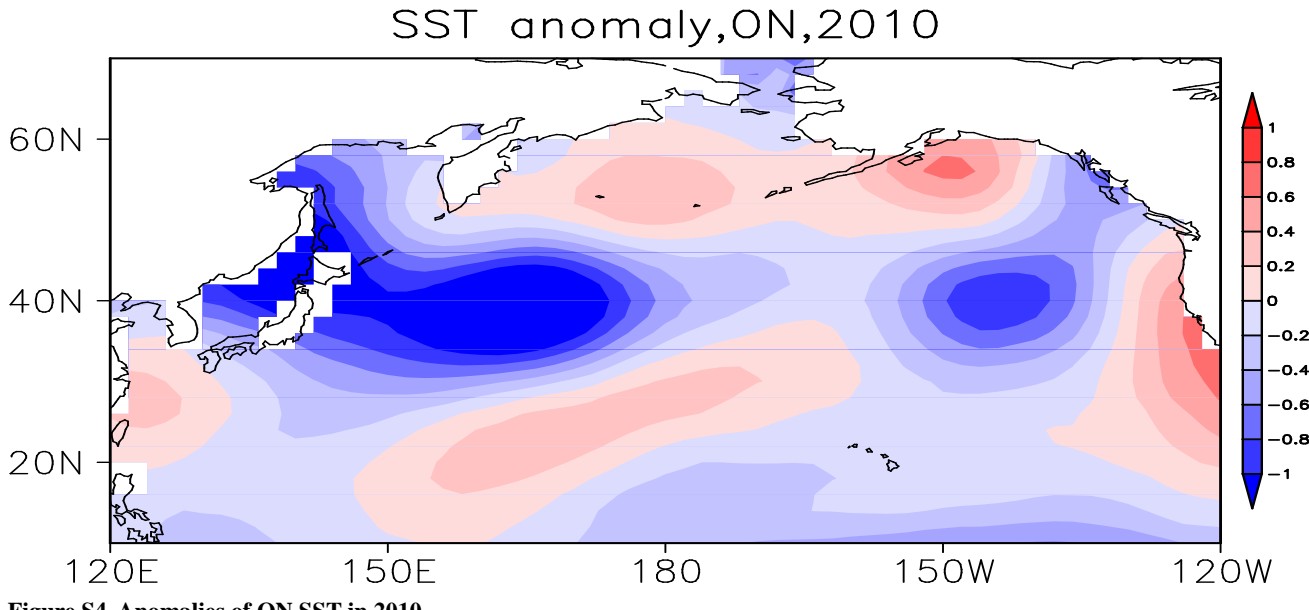

**520** **Figure S4. Anomalies of ON SST in 2010**

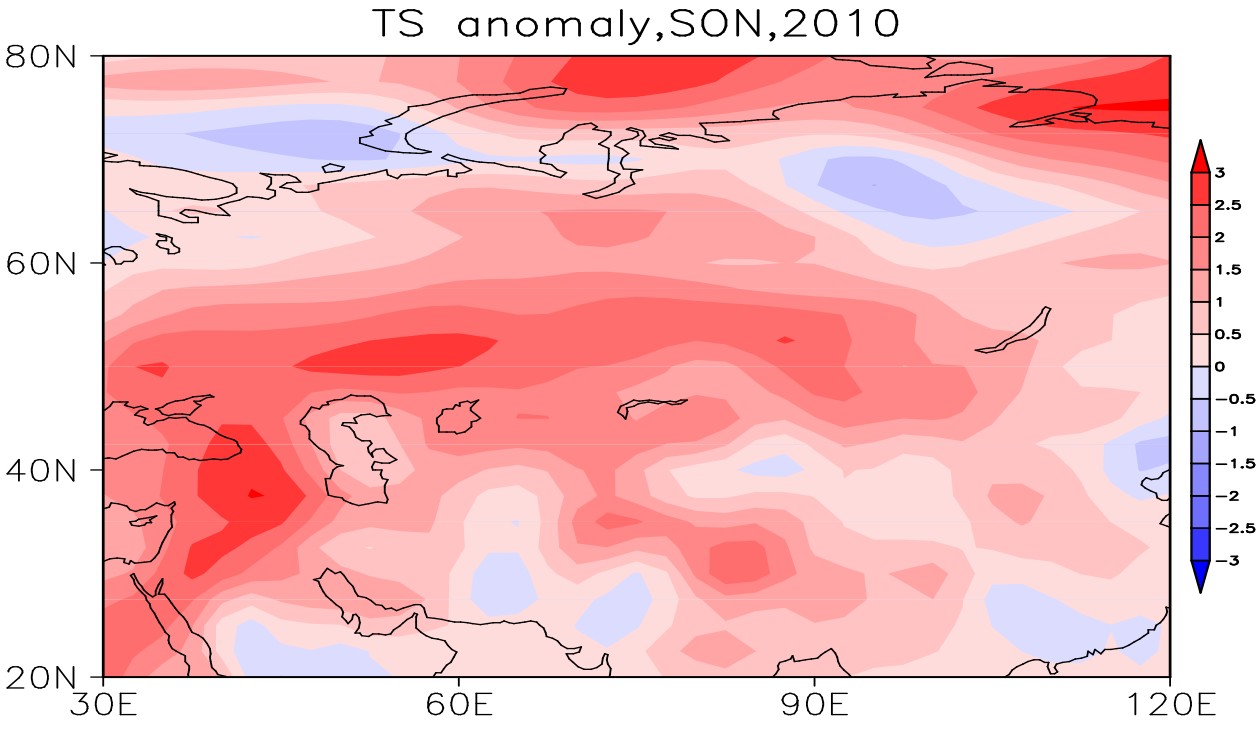

**Figure S5. Anomalies of pre-autumn TS in 2010**

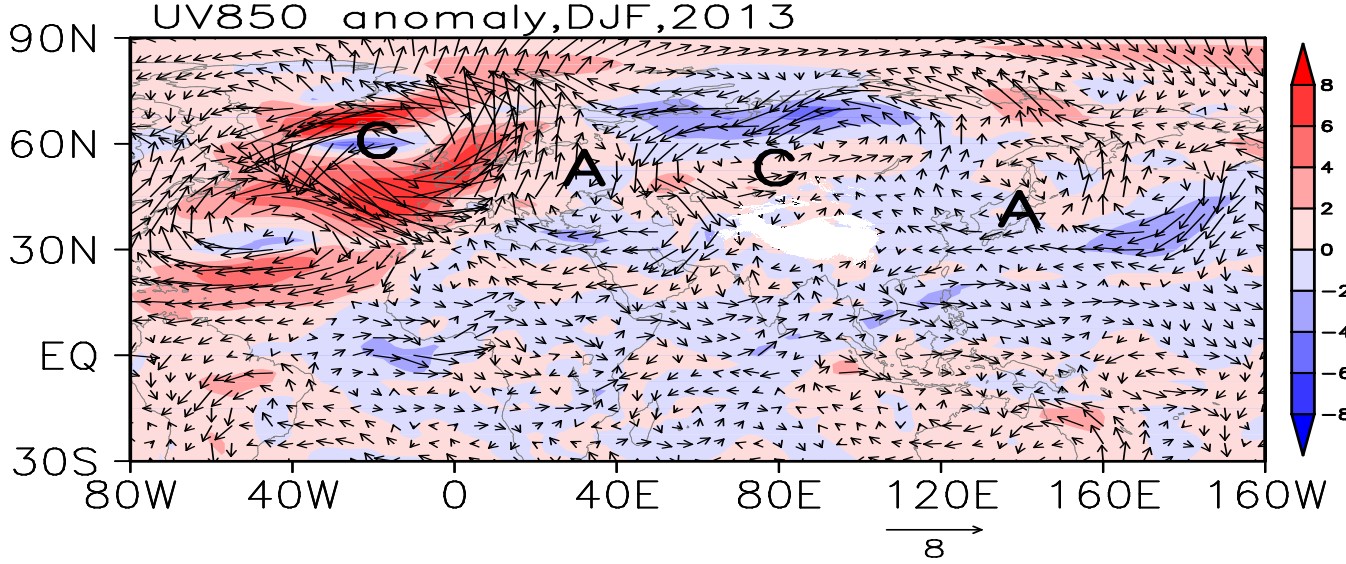

**Figure S6. UV850 (arrow) and speed (shade) anomalies in 2013. A and C represent anti-cyclone and cyclone, respectively.**

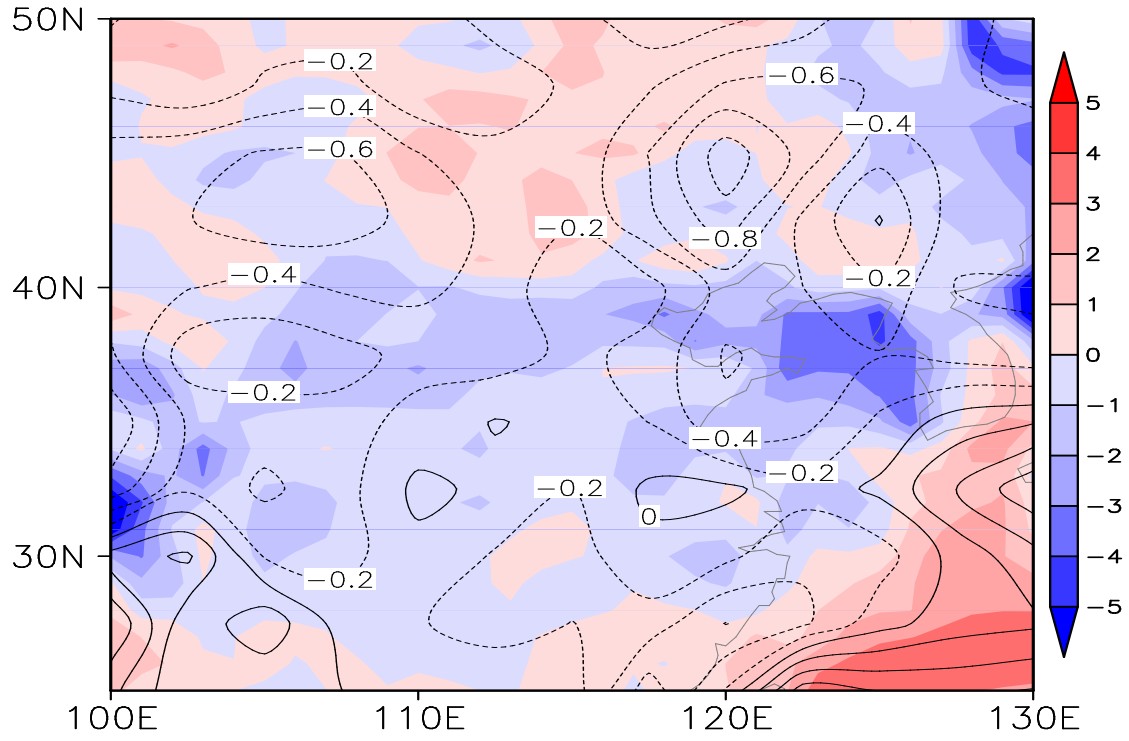

**Figure S7. Anomalies of surface wind speed (contour) and PBLH (shade) in winter 2013**

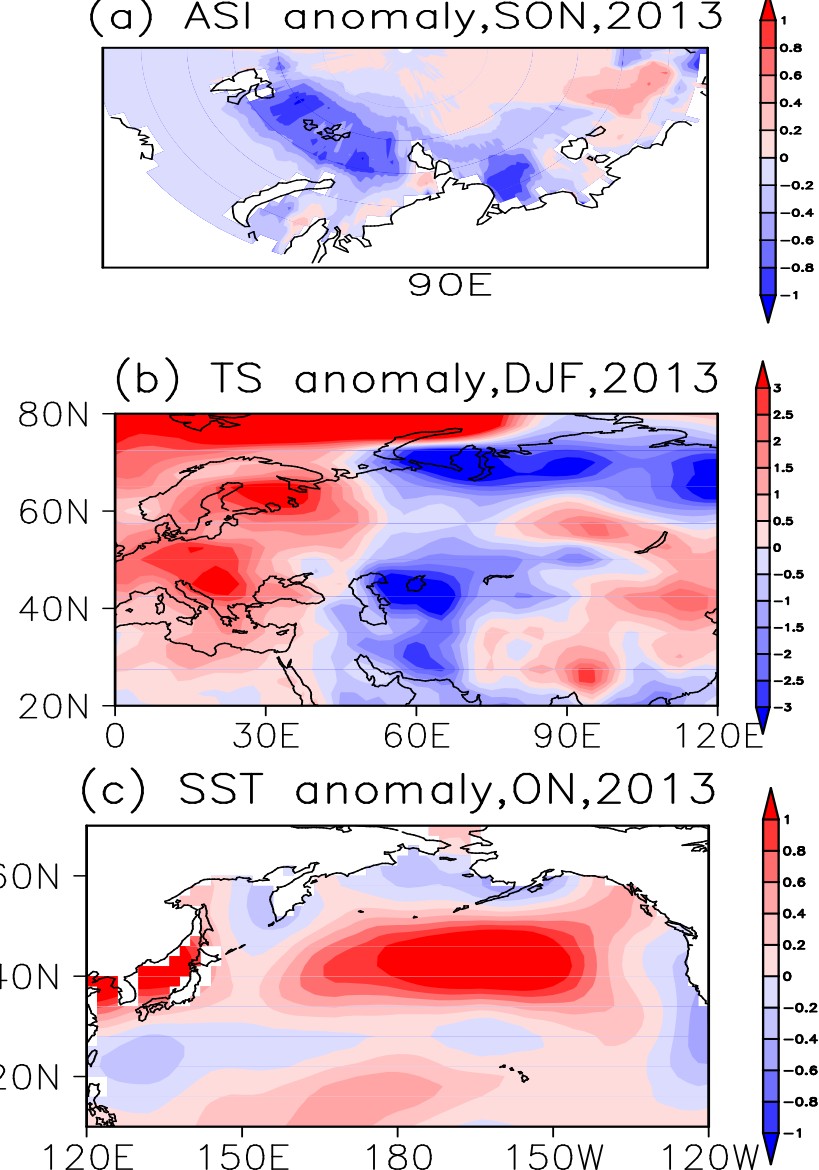

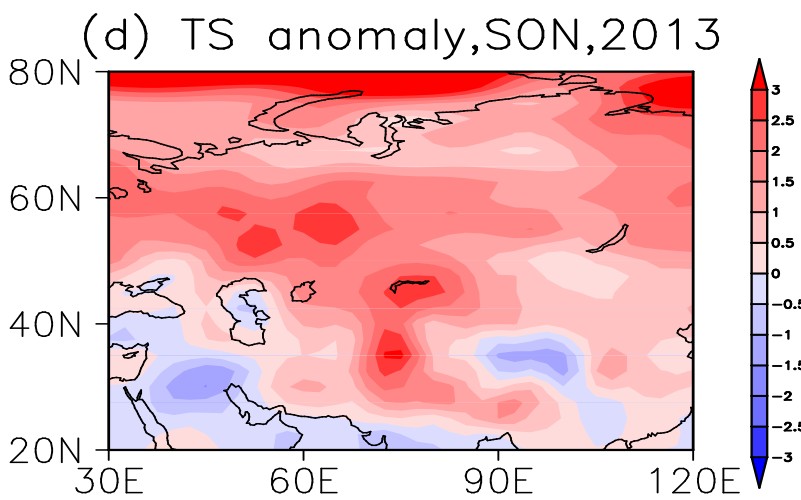

**Figure S8. Anomalies of external forcings in 2013. (a) ASI in pre-autumn, (b) TS in winter, (c) SST in Oct and Nov, and (d) TS in pre-autumn.**

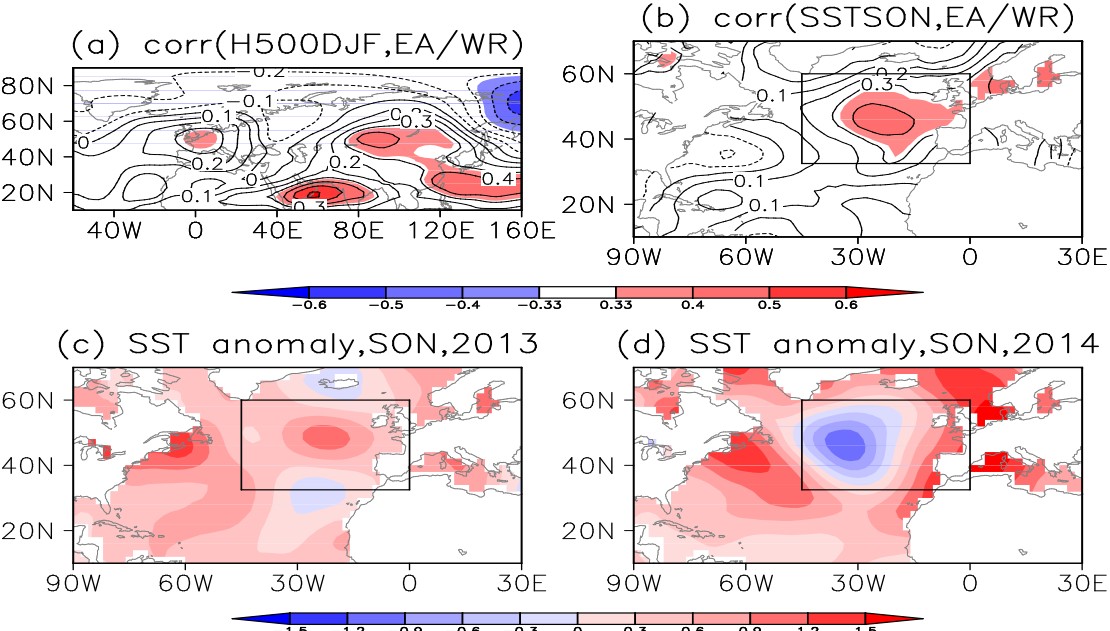

**Figure S9. Correlation coefficients between EA/WR index and H500 (a) / Atlantic SST in pre-autumn (b). Pre-autumn SST anomaly during SON in 2013 (c) and 2014 (d)**
