# Peer review of "Understanding Severe Winter Haze Events in the North China Plain in 2014: Roles of Climate Anomalies"

_Atmospheric Chemistry and Physics, 2016_

## Referee Comment (RC1) · Anonymous Referee #1 · 14 Sep 2016

The paper aimed at the severe haze events during the winter of 2014 happened in North-Central North China Plain, in which the authors attributed the main cause of haze in 2014 winter to the positive phases of the three teleconnection patterns: the East Atlantic/West Russia (EA/WR), the Western Pacific (WP), and the Eurasia (EU). The authors supplied some correlation analysis to explain the influences of three patterns to haze days over the North-Central North China Plain (WHDǎňNCP), and they put the results of SVD to illustrate the causes of three patterns in 2014. The severe haze events in China has become a hotspot in the research of atmospheric environment, thus the issue of this paper is interesting, as documented by some other papers in recent years. This paper pointed out the three patterns of teleconnection exerted important effects to haze days in China in 2014, and there are some other papers raised similar view of atmospheric circulation, therefore, there is no enough novelty in this

paper.

1, The WHDNCP shown in Fig. 4 includes significantly spatial changes in the research domain, and the domain is very small with respect to the spatial scales of three patterns. How can the authors explain the difference of WHDNCP in such a little region using so large three patterns?

2, Correlation coefficients have been supplied by in the paper to reveal the potential effects of three patterns to WHDNCP, but necessary physical analysis of such influence has been nearly ignored by the authors.

3, Concentration of PM2.5 reached its maximum in 2013, therefore the annual changes of emission should play an very important role in the haze days in recent years, but the author failed to arrange enough discussion and analysis about the emission change.

4, Line 20: EA/WA should be EA/WR

5, Line 36: What's the specific criterion of "static stability"?

6, Line 69-83: What's the role of Fig. 2 in such an analysis of large scale analysis for WHDNCP in 2014 ?

7, Line 109: two "that".

---

## Referee Comment (RC2) · Anonymous Referee #2 · 14 Oct 2016

In this work, the authors discussed the possible mechanisms of the severe haze pollution over the North China Plain in winter. Three teleconnection patterns (EA/WR, WP and EU) might have led to stable meteorological conditions that contributed to severe haze events over the North-Central North China Plain in 2014. Using SVD analysis, several external forcings were pointed out to have enhanced certain teleconnection patterns. This paper highlights the links between external forcings, teleconnection patterns and WHDs, but more discussions in detail were still needed in this paper:

1. 39 NCP stations were used to reconstruct the climatic WHDs, significant spatial variation of WHDs could be observed in Figure 4(a). Only four rural stations were selected to represent urban haze. The authors didn't introduce the locations of the four stations.

[Figure]

2. During the period from 1979 to 2012, the negative SLP anomaly in the Siberia region and the positive SLP anomaly over West Pacific led to weakened EAWM inducing the southeasterly anomaly (Figure 5 (a) and (c)), which was favorable for haze events. But in 2014, anomalies of meteorological fields (Figure 5 (b) and (d)) for WHDs in the NCP region were different from those in 1979-2012. Although the surface temperature was higher than average over the Asian continent, the surface wind fields over NCP region didn't show favorable conditions for WHDs. As anthropogenic emissions could also influence air quality, meteorological conditions might not be the main cause of WHDs in 2014.

3. Two extreme haze phenomenon were discussed in this work. Teleconnection patterns in 2010 and 2014 were different, but over the NCP region, anomalous circulations of wind fields in 2010 were similar with those in 2014. And the authors didn't discuss the regional meteorological conditions over NCP region in detail in 2010. Thus, in order to prove the importance of meteorological conditions on these two haze events the authors need to provide more evidences to support their arguments.

4. The authors concluded that anomalous circulations in winter 2013 were not as favorable for haze conditions as those in 2014. But the number of WHDs in 2013 was as large as that in 2014. Does it mean that the influence of anthropogenic influence was more significant in 2013 than in 2014? How could the authors eliminate the influence of anthropogenic influence?

5. In Line 10: inappropriate adjective "highest".
* * *

---

## Author Comment (AC1) · 15 Nov 2016

**Reply letter to the anonymous Referee #1**

The paper aimed at the severe haze events during the winter of 2014 happened in North-Central North China Plain, in which the authors attributed the main cause of haze in 2014 winter to the positive phases of the three teleconnection patterns: the East Atlantic/West Russia (EA/WR), the Western Pacific (WP), and the Eurasia (EU). The authors supplied some correlation analysis to explain the influences of three patterns to haze days over the North-Central North China Plain ($WHD_{NCP}$), and they put the results of SVD to illustrate the causes of three patterns in 2014. The severe haze events in China has become a hotspot in the research of atmospheric environment, thus the issue of this paper is interesting, as documented by some other papers in recent years. **This paper pointed out the three patterns of teleconnection exerted important effects to haze days in China in 2014, and there are some other papers raised similar view of atmospheric circulation, therefore, there is no enough novelty in this paper.**

*Reply***:**

Since several years ago, we have been working on the impact of climate change on the haze pollution in China (associated publications were listed below) and keeping tracking the related researches. We confirmed that this study had some new and key progress, such as:

(1) We investigated the joint effects of the EA/WR, EU and WP teleconnection patterns for the first time and pointed out the pivotal role of the anomalous high over North China.

(2) The latest extremes of haze days in 2010, 2013 and 2014 were analyzed. Not from a specific perspective, but the comprehensive climate conditions (i.e., the related anomalous circulations and external forcings) were researched.

(3) By comparing 2010 and 2014, we confirmed both local climate conditions and climate teleconnections influenced haze weather very much.

(4) Furthermore, the reconstructed haze dataset based on meteorological observations

agree well with the atmospheric composition measurements (particularly the GAWS), thus providing a basis for long-term variability study on the haze pollution.

*Associated publications*

1. Chen H. P., and **H. J. Wang**, 2015: Haze days in North China and the associated atmospheric circulations based on daily visibility data from 1960 to 2012. J. Geophys. Res. Atmos., 120, 5895-5909, doi:10.1002/2015JD023225.

2. **Wang H. J.**, H. P. Chen, and J. Liu, 2015: Arctic sea ice decline intensified haze pollution in eastern China. Atmos. Oceanic Sci. Lett., 8, 1-9.

3. **Wang H. J.**, H. P. Chen. 2016: Understanding the recent trend of haze pollution in eastern China: roles of cliamte change. Atmos. Chem. Phys., 16, 4205—4211

4. **Yin Z. C., H. J. Wang**, 2015: The relationship between the subtropical Western Pacific SST and haze over North-Central North China Plain. Int. J. Climatol., doi:10.1002/joc.4570.

5. **Yin Z. C., H. J. Wang,** W. L. Guo, 2015: Climatic change features of fog and haze in winter over North China and Huang-Huai Area. SCIENCE CHINA EarthSciences, 58(8): 1370-1376.

6. **Yin Z. C., H. J. Wang,** D. M. Yuan, 2015: Interdecadal increase of haze in winterover North China and the Huang-huai Area and the weakening of the East Asia Winter Monsoon,Chin Sci Bull, 2015, 60: 1395–1400 (in Chinese).

7. **Yin, Z. and Wang, H.**: Seasonal Prediction of Winter Haze Days in the North-Central North China Plain, Atmos. Chem. Phys., doi:10.5194/acp-2016-691, accetped, 2016.

**1, The WHD$_{NCP}$ shown in Fig. 4 includes significantly spatial changes in the research domain, and the domain is very small with respect to the spatial scales of three patterns. How can the authors explain the difference of WHD$_{NCP}$ in such a little region using so large three patterns?**

*Reply*:

**Due to the influence of human activities, the distribution of WHD shows significantly spatial changes among the rural area, small cities and metropolises.** In the paper, we supplemented some analysis about the rural stations in Figure 1. From the Figure 1 and 2a, we can see that the areas with less WHD were near the rural stations. Actually, the increase of WHD in rural area was an obvious reflection of the severe haze disaster in recent years. In other words, **the coverage of haze invaded into the rural region in 2014.**

We found that l**arge scale patterns could impact the haze events in the relatively small region by altering the local meteorological conditions via teleconnection** (i.e., EA/WR teleconnection pattern, WP teleconnection pattern and EU teleconnection pattern). For example, when the WP pattern showed stronger positive phase, there were stronger positive anomalies of geopotential height on surface, 850 hPa and 500 hPa over North China and northwest Pacific. The anomalous anti-cyclone was enhanced, thus, the vertical motion or convection on the NCP was confined and the southerly anomalies on the left side of the anti-cyclone were induced to weaken the cold air and wind speed. This is the way that the WP teleconnection pattern altered the local meteorological conditions. Under such meteorological conditions, the vertical and horizontal diffusions of atmospheric particulates were both restricted, and then the pollutant gathered in a narrow space that resulted in the occurrence of haze. **The main process could be summarized as "climate teleconnection → local climate conditions → weaker diffusion conditions → pollutant gathered in a narrow space".** The physical process related with EA/WR and EU were similar to that was described above. The Figures 4—5 was revised and the order was exchanged to match the improvement of the physical analysis. The revised text and Figures was showed in the replies to "Specific comment 2".

Additionally, the impact of ENSO (Gao et al. 2015) and Tibetan Plateau (Xu et al. 2016) on the haze frequency in eastern China was also obvious, whose scales were even larger. What's more, the North China severe summer drought in 2014 also been

explained by the large scale teleconnections (Wang et al. 2015).

*Associated references*

Wang H J, He S P. 2015. The North China/Northeastern Asia Severe Summer Drought in 2014. Journal of Climate. 28(17), 6667–668

Gao H, Li X. 2015. Influences of El Nino Southern Oscillation events on haze frequency in eastern China during boreal winters. International Journal of Climatology. 35

Xu X et al. 2016. Climate modulation of the Tibetan Plateau on haze in China. Atmospheric Chemistry and Physics, 16(3): 1365-1375

***Revision:***

……Due to the quality and temporal range of the data, only four rural stations were qualified and selected (white circles in Figure 1)…….

[Figure]

Figure 1. Topographic map (shading; unit: m) of North China and the locations of 39 NCP observation sites (Urban: black circle, Rural: white circle). The NCP area is represented by a black rectangle, and the names of provinces and mountains are written in red and white, respectively.

…… As shown in Figure 1, there were four rural stations, three of which were located near the Yan Mountains and were corresponding to less WHD. Another rural site was near the boundaries of Shandong and Henan (BSH) and also resulted in less

WHD. Figure 3b shows the WHD anomalies in 2014 with respect to 1979–2012. In addition to a few sites, a larger number of WHD occurred, especially on the BSH (rural area) and the northeast of Hebei. It is notable that WHDs in these two regions show significant increases, filling up the climatic WHD valley as shown in Figure 3a. As a result, the haze-prone area joined together, indicating that the haze pollution was more serious in this region. Actually, the fast increase of WHD in rural area was an obvious reflection of the severe haze disaster in recent years.…….

**2, Correlation coefficients have been supplied by in the paper to reveal the potential effects of three patterns to WHD$_{NCP}$, but necessary physical analysis of such influence has been nearly ignored by the authors.**

*Reply*:

Some necessary physical analysis has been added in section 3. The main physical mechanism was that **"climate teleconnection → local climate conditions → weaker diffusion conditions → pollutant gathered in a narrow space"**. The Figures was revised and **the order was exchanged to match the description of the physical analysis (from teleconnection to local conditions)**. The detailed analysis and revision was listed below.

*Revision in section 3:*

[revised manuscript text omitted]

**3, Concentration of PM2.5 reached its maximum in 2013, therefore the annual changes of emission should play an very important role in the haze days in recent years, but the author failed to arrange enough discussion and analysis about the emission change.**

*Reply*:

**To enhance our perspective, the title has been changed to "Understanding Severe Winter Haze Events in the North China Plain in 2014: Roles of Climate Anomalies".** The main purpose of this study was to discuss the roles of climate conditions. **The impact of emissions was written in the "Introduction" and "Discussion".**

No doubt that the long-term increase of pollutant emission is the fundamental factor for the haze pollution enhancement in recent years. However, so far, **there is no evidence that the emissions were more in 2014 than in 2010.** Furthermore, the $PM_{2.5}$ concentration of a global atmospheric watch station (Shangdianzi) and an urban station (Baolian) were **almost equal** in winter 2010 and 2014. By our current and some previous analysis, we found that **the climate factor played key roles in the formation of the heavy pollution case like 2014**.

[Figure]

Figure 11. Mean mass concentration of $PM_{2.5}$ in winter at Shangdianzi (red; measured by the TOEM (solid) and β-ray (red dash) method) and Baolian (black) Station. The error bar represents one standard error among the different measured hours.

The anthropogenic emissions were the fundamental driver and mostly impacted the long-term trend of the haze pollution. To some extent, the energy consumption

varied continuously and linearly in eastern China and **the socio-economic components of WHD$_{NCP}$ could be removed primitively by detrending**, and then the interannual variability of haze pollution should be mainly the result of climatic anomalies.

The case studies of 2013 enhanced that the pollutant emission is the fundamental factor and highlighted the roles of anthropogenic influence. **
[revised manuscript text omitted]

**5, Line 36: What's the specific criterion of "static stability"?**

*Reply***:**

The "static stability" was the ability of a fluid at rest to become turbulent or laminar due to the effects of buoyancy. The static stability of the atmosphere could indicate the vertical motion potential (or the intensity of convection) that was important for weather and pollution forecast.

Throughout the paper, the term "static stability" was only used once and in the introduction. Thus, we decided to correct it to a clearer and simpler expression "stable atmosphere".

*Revision:*

SWP-SST weakened EAWM circulation, leading to a favorable environment for haze with stable atmosphere and potential for hygroscopic growth (Yin et al. 2016).

**6, Line 69-83: What's the role of Fig. 2 in such an analysis of large scale analysis for WHD$_{NCP}$ in 2014 ?**

*Reply***:**

As we know, haze is a multidisciplinary phenomenon that can be represented by visibility and humidity in meteorology, and by the concentration of the atmospheric composition in environmental science. The older Figure 2 was plotted to demonstrate

the representation of haze data reconstructed primarily by visibility. After comparison, we confirmed that **the derived haze datasets not only agreed with the meteorological standard but also satisfactorily represented the concentration of the atmospheric composition.**

Following the reviewers advice, we putted this Figure into the supplementary material (i.e., Figure S1).

*Revision:*

Figure S1. Visibility of Beijing (green) and atmospheric………

**7, Line 109: two "that".**

*Reply***:**

The error has been corrected.

*Revision:*

It is notable that WHDs in these two regions show significant increases, filling up the climatic WHD valley as shown in Figure 3a.

---

## Author Comment (AC2) · 15 Nov 2016

**Reply letter to the anonymous Referee #2**

In this work, the authors discussed the possible mechanisms of the severe haze pollution over the North China Plain in winter. Three teleconnection patterns (EA/WR, WP and EU) might have led to stable meteorological conditions that contributed to severe haze events over the North-Central North China Plain in 2014. Using SVD analysis, several external forcings were pointed out to have enhanced certain teleconnection patterns. This paper highlights the links between external forcings, teleconnection patterns and WHDs, but more discussions in detail were still needed in this paper:

**1. 39 NCP stations were used to reconstruct the climatic WHDs, significant spatial variation of WHDs could be observed in Figure 4(a). Only four rural stations were selected to represent urban haze. The authors didn't introduce the locations of the four stations.**

*Reply*:

 **Our motivation is to address the haze pollution in large area, including both the urban and rural regions.** The reason for that only 4 rural sites were selected was the limited quality and temporal range of the rural measurements. However, reminded by the reviewer, **the Figure 1 was revised to introduce the location of the rural sites and some new features were revealed.** From the Figure 1 and 2a, we can see that the areas with less WHD were near the rural stations. Actually, the fast increase of WHD in rural area was an obvious reflection of the severe haze disaster in recent years. In other words, the coverage of haze invaded into the rural region in 2014.

*Revision:*

 ……Due to the quality and temporal range of the data, only four rural stations were qualified and selected (white circles in Figure 1).  ……

[Figure]

**Figure 1. Topographic map (shading; unit: m) of North China and the locations of 39 NCP observation sites (Urban: black circle, Rural: white circle). The NCP area is represented by a black rectangle, and the names of provinces and mountains are written in red and white, respectively.**

……As shown in Figure 1, there were four rural stations, three of which were located near the Yan Mountains and were corresponding to less WHD. Another rural site was near the boundaries of Shandong and Henan (BSH) and also resulted in less WHD. Figure 3b shows the WHD anomalies in 2014 with respect to 1979−2012. In addition to a few sites, a larger number of WHD occurred, especially on the BSH (rural area) and the northeast of Hebei. It is notable that WHDs in these two regions show significant increases, filling up the climatic WHD valley as shown in Figure 3a. As a result, the haze-prone area joined together, indicating that the haze pollution was more serious in this region. Actually, the fast increase of WHD in rural area was an obvious reflection of the severe haze disaster in recent years.…….

**2. During the period from 1979 to 2012, the negative SLP anomaly in the Siberia region and the positive SLP anomaly over West Pacific led to weakened EAWM inducing the southeasterly anomaly (Figure 5 (a) and (c)), which was favorable for haze events. But in 2014, anomalies of meteorological fields (Figure 5 (b) and (d)) for WHDs in the NCP region were different from those in 1979-2012.**

Although the surface temperature was higher than average over the Asian continent, the surface wind fields over NCP region didn't show favorable conditions for WHDs. As anthropogenic emissions could also influence air quality, meteorological conditions might not be the main cause of WHDs in 2014.

*Reply*:

(1) So far, **there is no evidence that the emissions were more in 2014 than 2010**. Furthermore, the $PM_{2.5}$ concentration of a global atmospheric watch station (Shangdianzi) and an urban station (Baolian) were **almost the same** in winter 2010 and 2014. By our current and some previous analysis, we found that **the climate factor played key roles in the formation of the heavy pollution case like 2014**.

The anthropogenic emissions were the fundamental driver and mostly impacted the long-term trend of the haze pollution. To some extent, the energy consumption varied continuously and linearly in eastern China and **the socio-economic components of $WHD_{NCP}$ could be removed primitively by detrending**, and then the interannual variability of haze pollution should be mainly the result of climatic anomalies.

[Figure]

Figure 11. Mean mass concentration of $PM_{2.5}$ in winter at Shangdianzi (red; measured by the TOEM (solid) and β-ray (red dash) method) and Baolian (black) Station. The error bar represents one standard error among the different measured hours.

To enhance our perspective, the title has been changed to "Understanding Severe Winter Haze Events in the North China Plain in 2014: Roles of Climate Anomalies". The main purpose of this study was to discuss the roles of climate anomalies and the impact of emissions is discussed in the "Introduction" and "Discussion".

(2) In the discussion section, we pointed out "the associated circulations and external forcings in 2010 were still slightly different from those in 2014. It is possible that **not all of the above factors might be found in a specific case study**, i.e., **a few of these factors played the essential roles and led to the characteristics of that case**. A brief summary of the impacts of these factors on $WHD_{NCP}$ is offered in Table 2". The Table 2 was improved to include all of the possible factors. In summary, the local climate conditions, (i.e., weaker surface wind speed and lower PBLH), the teleconnection patterns and the external forcings were benefit for the occurrence of the haze extreme in 2014.

Table 2. Summary of the various influnce factors for $WHD_{NCP}$. The "+++" indicates "more important"; "++" indicates "important", "+" indicates "less important" , and blank indicates "not important".

| Factors | 2010 | 2013 | 2014 |
|---|---|---|---|
| $PM_{2.5}$ concentration | ++ | +++ | ++ |
| Local surface wind speed | ++ | ++ | ++ |
| Local PBLH | ++ | ++ | ++ |
| EA/WR | ++ | ++ | ++ |
| WP | ++ | | ++ |
| EU | | | ++ |
| Pre-autumn ASI | ++ | | ++ |
| Winter TS | ++ | | ++ |
| ON Pacific SSTA | + | ++ | ++ |
| Pre-autumn TS | ++ | | ++ |

To enhance our analysis of 2014 case, the anomalies of meteorological fields were re-plotted and the composite of PBLH was supplemented. ① **For the horizontal diffusion**: The main argument was that the weaker EAWM pattern could intensify the haze pollution in the NCP. Except the SLP anomalies over Northeast

China and Japan Sea, the pressure field was also similar with the climatic correlation. Furthermore, the surface wind anomalies at the high latitude blocked the cold air from their source and there were southerly to the south of $40^{\circ}$N that occupied most of our research area. Unfortunately, the isolated northerly could be found to the north of $40^{\circ}$N. That is, the large scale EAWM pattern and most of the synoptic scale circulations were benefit for the large number of $WHD_{NCP}$ in winter 2014. ② **For the vertical diffusion**: The composite of PBLH was supplemented as Figure 6. The PBLH was a good indicator for air pollution, especially for the vertical diffusivity. The winter PBLH in 2014 was much lower than that in 2010, so the vertical diffusivity was worse in 2014 resulting in more haze days.

*Revisions to the impact of human activities were showed in the replies to Q4:*

*Revision to the meteorological conditions in 2014:*

......Nevertheless, the three eastern centers of EA/WR, WP and EU patterns could be recognized (Figure 4c). The linkage anti-cyclone of these three teleconnection patterns was enhanced and modulated the local climate conditions. The NCP area was influenced by the anomalous high resulting in lower PBLH (Figure 6). The southerly at the high latitudes deadened the cold air from its main source, so the atmospheric matters gathered easily. Near the surface, the negative SLP anomalies occupied the whole Asian continent and Japan Sea that weakened the continental cold high and stimulated significant southerly anomalies to the north of $50^{\circ}$N with the weaker Aleutian low. The weaker EAWM circulations near the surface blocked the cold air from high latitudes and resulted in warmer surface (Wang et al. 2015). There were positive SLP anomalies over South China Sea and East China Sea that induced southerly and smaller surface wind speed over the coastal area in the east of China (Figure 5c—d). The diffusivity of atmosphere over the NCP was limited, so the high pollutant emissions were concentrated in a narrow space and severe haze events occurred easily..……

[Figure]

**Figure 5.** Correlation coefficients between WHD$_{NCP}$ and winter circulations from 1979 to 2012 with linear trend was removed (a, b), and circulation anomalies in 2014 (c, d) and 2010 (e, f). The circulations in (a, c, e) are TS (shade) and SLP (contour) and those in (c, d, f) are surface wind speed (shade) and wind vector (arrow).

[Figure]

**Figure 6.** The difference of winter PBLH between 2014 and 2010

**3. Two extreme haze phenomenon were discussed in this work. Teleconnection patterns in 2010 and 2014 were different, but over the NCP region, anomalous circulations of wind fields in 2010 were similar with those in 2014. And the authors didn't discuss the regional meteorological conditions over NCP region in detail in 2010. Thus, in order to prove the importance of meteorological conditions on these two haze events the authors need to provide more evidences to support their arguments.**

*Reply***:**

**The regional meteorological conditions in 2010 were supplemented as Figure 5e—f and 6 that showed similar but opposite features**. As reply to Q2, we discussed that it is possible that not all of the above factors might be found in a specific case study, i.e., only a few of these effective factors **played the essential roles and** led to the characteristics of that case. A brief summary of the impacts of these factors on $WHD_{NCP}$ is offered in Table 2". ① **For 2010**, the local climate conditions (i.e., weaker surface wind speed and lower PBLH), most of the teleconnections (except the EU) and the external forcings played positive roles to less haze days. ② **For 2014**, In the local climate conditions, (i.e., weaker surface wind speed and lower PBLH), the teleconnection patterns and the external forcings were benefit for the occurrence of the haze extreme in 2014.

**Table 2. Summary of the various influnce factors for $WHD_{NCP}$. The "+++" indicates "more important"; "++" indicates "important", "+" indicates "less important" , and blank indicates "not important".**

| Factors | 2010 | 2013 | 2014 |
|---|---|---|---|
| $PM_{2.5}$ concentration | ++ | +++ | ++ |
| Local surface wind speed | ++ | ++ | ++ |
| Local PBLH | ++ | ++ | ++ |
| EA/WR | ++ | ++ | ++ |
| WP | ++ | | ++ |
| EU | | | ++ |
| Pre-autumn ASI | ++ | | ++ |
| Winter TS | ++ | | ++ |
| ON Pacific SSTA | + | ++ | ++ |
| Pre-autumn TS | ++ | | ++ |

*Revision:*

……Large scale circulations, such as the negative phase of EA/WR and WP, were quite clear in 2010 (Figure 4d). Near the surface, the anomalous circulations were distributed similarly but almost opposite, i.e., the stronger continental cold high and oceanic low (Figure 5e), the northerly and stronger surface wind over NCP

(Figure 5f), and the higher PBLH (Figure 6). The atmospheric diffusivity was heightened by the stronger cold air and vertical movement………

[Figure]

**Figure 5. Correlation coefficients between WHD$_{NCP}$ and winter circulations from 1979 to 2012 with linear trend was removed (a, b), and circulation anomalies in 2014 (c, d) and 2010 (e, f). The circulations in (a, c, e) are TS (shade) and SLP (contour) and those in (c, d, f) are surface wind speed (shade) and wind vector (arrow).**

**4. The authors concluded that anomalous circulations in winter 2013 were not as**

**favorable for haze conditions as those in 2014. But the number of WHDs in 2013 was as large as that in 2014. Does it mean that the influence of anthropogenic influence was more significant in 2013 than in 2014? How could the authors eliminate the influence of anthropogenic influence?**

*Reply***:**

(1) No doubt that the long-term increase of pollutant emission is the fundamental factor for the haze pollution enhancement in recent years. **However, so far, there is no evidence that the emissions were more in 2014 than in 2010.** By our current and some previous analysis, we found that the climate factor played key roles in the formation of the heavy pollution case like 2014. After removal of the linear trend, the interannual variability of haze pollution should be mainly the result of climatic anomalies. **Thus, we removed the socio-economic components of WHD$_{NCP}$ primitively by detrending.** To enhance our perspective, the title has been changed to "Understanding Severe Winter Haze Events in the North China Plain in 2014: Roles of Climate Anomalies".

(2) The impact of emissions was discussed in the "Introduction" and "Discussion". **We also talked about the anthropogenic influence to the 2013 case. The concentration in winter 2013 increased abruptly up to nearly twice that in 2010 and 2014 and was the highest in the observation history that broke down our assumption.** Even the anomalous circulations were not benefit enough for haze occurrence, the joint effect of highest pollution emissions and climate conditions could result in the serious haze event.

In the former version, we concluded that the anomalous in 2013 was not favorable as 2014 curtly. Now, we decided to cancel this conclusion and discuss the comparison more objectively in the last section to show the difference between 2013 and 2014.

[revised manuscript text omitted]